

# U-Th and [10]Be constraints on sediment recycling in proglacial settings, Lago Buenos Aires, Patagonia

Antoine Cogez[1], Frédéric Herman[1], Éric Pelt[2], Thierry Reuschlé[3], Gilles Morvan[2], Christopher M. Darvill[4], Kevin P. Norton[5], Marcus Christl[6], Lena Märki[1], and François Chabaux[2]

[1]University of Lausanne, Institut des Dynamiques de la Surface Terrestre, Quartier Unil-Mouline, Bâtiment Geopolis, CH-1015 Lausanne, Switzerland
[2]University of Strabourg, Laboratoire d'Hydrologie et de Géochimie de Strasbourg, Strasbourg, France
[3]University of Strabourg, Institut de Physique du Globe de Strasbourg, Strasbourg, France
[4]Geography Program and Natural Resources and Environmental Studies Institute, University of Northern British Columbia, 3333 University Way, Prince George, BC, V2N 4Z9, Canada
[5]Victoria Univeristy of Wellington, School of Geography, Environment, and Earth Sciences, Wellington, New Zealand
[6]Swiss Federal Institute of Zürich, Ion Beam Physics Laboratory, Zürich, Switzerland

*Correspondence to:* Antoine Cogez (antoine.cogez@gmail.com)

**Abstract.** The estimation of sediment transfer times remains a challenge to our understanding of sediment budgets and the relationships between erosion and climate. Uranium (U) and Thorium (Th) isotope disequilibria offer a means of more robustly constraining sediment transfer times. Here, we present new Uranium and Thorium disequilibrium data for a series of nested moraines around Lago Buenos Aires in Argentine Patagonia, as well as a refined glacial chronology for the area using *in situ* cosmogenic [10]Be analysis from corresponding glacial outwash. Sediment transfer times within the periglacial domain were estimated by comparing the deposition ages of moraines to the theoretical age of sediment production, i.e. the comminution age inferred from U disequilibrium data and recoil loss factor estimates. Our data show first that the classical comminution age approach must include weathering processes, accounted for by measuring Th disequilibrium. Second, our combined data suggest that the pre-deposition history of the moraine sediments is not negligible, as evidenced by the large disequilibrium of the youngest moraines despite the equilibrium of corresponding glacial flour. Monte Carlo simulations suggest that weathering was more intense before the deposition of the moraines and that the transfer time of the fine sediments to the moraines was on the order of 100-200 ka. Long transfer times could result from a combination of long sediment residence times



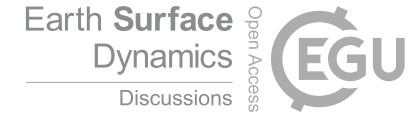

in the proglacial lake (recurrence time of a glacial cyle) and the remobilization of moraines deposited during previous glacial cycles. [10]Be data suggest that some glacial cycles are absent from the preserved moraine record (seemingly every second cycle), supporting a model of reworking moraines and/or fluctuations in the extent of glacial advances. The chronological pattern is consistent with the U-Th disequilibrium data and the 100-200 ka transfer time. This long transfer time raises the question of the proportion of freshly eroded sediments that escape or not the proglacial environments during glacial periods.

*Copyright statement.* TEXT

# 1 Introduction

The sedimentary cycle incorporates the erosion of rock followed by the transport and deposition of sediment. While rates of erosion and deposition can be accurately documented, tracing the history of sediments between production and deposition remains challenging. Importantly, mechanisms of transfer and alteration of sediments during transport play a key role in the evolution of basins and feedbacks between erosion and climate, especially because the age of sediment strongly controls its susceptibility to weathering (e.g. White & Brantley, 2003; Vance et al., 2009). This is particularly the case in glacial settings because glaciers are highly efficient at eroding landscapes (e.g. Hallet et al., 1996; Koppes & Montgomery, 2009) and produce highly-reactive and easily-weathered sediments (Anderson et al., 1997; White & Brantley, 2003; Anderson, 2005). Moreover, glaciers can create large overdeepenings that are subsequently filled with sediments isolated from interactions with surface processes. The efficiency of evacuating those sediments post-deposition is largely unknown.

Silicate weathering is an important surface parameter for cooling climate over geologic and glacial-interglacial timescales because it consumes $CO_2$ (Ebelmen; Walker et al., 1981). But its role in controlling $CO_2$ concentrations and climate variations over glacial-interglacial cycles is contentious (Foster & Vance, 2006; Vance et al., 2009; Lupker et al., 2013; VonBlanckenburg et al., 2015; Cogez et al., 2015). Understanding whether weathering varied over these cycles requires robust determinations of sediment transfer times.



For example, long transport times can cause a lag and/or a damping of the response of weathering to climate or erosion forcing and bias reconstructions of erosion and weathering intensity and variations through time.

Geochemical tools offer a means of measuring sediment transport times. Uranium series isotopes are particularly useful because the diversity of chemical elements in the radioactive decay chain (U, Th and
5 Ra) leads to disequilibrium during surface process fractionation. Moreover, the half-lives of these isotopes range from 1,500 to 250,000 years, corresponding to sediment transport process times. The timescales of weathering processes have been particularly well documented using U-Th-Ra disequilibria in soil sediments and river waters (e.g. Ackerer et al., 2016; Chabaux et al., 2008, 2012, 2013; Dosseto & Schaller, 2016; Dosseto et al., 2010, 2012, 2014; Granet et al., 2010; Keech et al., 2013; Ma et al., 2013). An alternative
approach uses the fine fraction of silicates to date the time since physical erosion of sediments (DePaolo et al., 2006, 2012). The method is based on the $\alpha$-recoil of uranium during radioactive decay, triggering the loss of a fraction of the daughter isotopes compared to the parent in small ($\leq 50\mu$m) grains. The time since comminution can theoretically be estimated, despite difficulties and limitations discussed in previous studies (Maher et al., 2006; Lee et al., 2010; Handley et al., 2013a, b; Dosseto et al., 2010). However, this method
only involves the $^{234}$U/$^{238}$U ratio and so neglects the effect of chemical weathering and further reduction of grain size after comminution.

In this study, we sampled sequences of nested moraines around Lago Buenos Aires in Argentine Patagonia, which range in age from 0 to 1 Ma (Singer et al., 2004; Kaplan et al., 2005, e.g.). The aim was to constrain the pre-deposition history of sediments within the moraines. First we refined the deposition age
of five of the moraines using *in situ* cosmogenic $^{10}$Be exposure and depth profile dating. To constrain the sediment transport times, we used the comminution ages approach combined with $^{230}$Th disequilibrium measurements. We show that taking weathering into account can help to resolve previous issues with the comminution ages methodology. Finally, we are able to estimate that the pre-deposition sediment history is likely on the order of 100-200 ka, including sediment recycling and chemical weathering in the proglacial
system. Our findings have implications for the understanding of the couplings between erosion and climate.

## 2 Settings and methods

### 2.1 The Lago Buenos Aires moraines

Lago Buenos Aires in Argentina (Lago General Carreras in Chile) is a large proglacial lake, around 100 km long and 5-20 km wide, oriented east-west, at 46°S (Figure 1). The present climate of the area is temperate,

with an average annual precipitation of 100 mm/yr and temperatures between 4 and 14 °C, producing open steppe vegetation. On the eastern edge of the lake, a series of frontal moraines are nested from the youngest close to the lake to the oldest 50 km further east. The chronology of these moraines has been already studied: the five innermost moraines (Fenix 1-5) are Last Glacial Maximum (LGM) in age (Kaplan et al., 2004; Douglass et al., 2006).Singer et al. (2004) dated lava flows interbedded with six of the intermediary moraines

(Moreno 1-3 and Deseado 1-3) and showed that they range in age between 109 ka and 760 ka. Finally, six moraines (Telken 1-6) in the outermost part of the system were likely deposited between 760 ka and 1016 ka. Cosmogenic nuclide exposure dating of boulders supports the assertion that erosion and degradation of older moraines in this region yields erroneously young ages (e.g. Kaplan et al., 2005) Hein et al. (2009). Recently, Hein et al. (2017) used exposure dating of cobbles on outwash related to the Moreno moraines to

show that they were constructed at ca. 260-270 ka, during Marine Isotope Stage (MIS) 8. For this study, we further refined the chronology by providing direct age constraints for the Deseado 1, Deseado 2 and Telken 5 moraines (see section 2.2).

U and Th samples were taken from silty beds inside the moraines, several meters below the surface in order to avoid potential post depositional modifications (such as weathering or dusts inputs). In addition to

moraine samples, a glacial flour sample was taken at the front of the Los Exploradores Glacier in Laguna San Rafael National Park to estimate the initial U-Th composition of silts found in the moraines (Figure 1). This glacier covers part of the total catchment draining into Lago Buenos Aires. Consequently, it is unlikely to be representative of the entire basin but provides an estimate of the initial composition of silts after comminution.

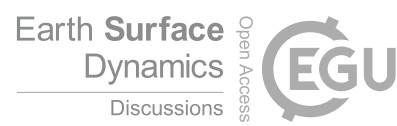

## 2.2 [10]Be methodology

### 2.2.1 Sampling

Our cosmogenic nuclide exposure dating methodology followed the sampling recommendations of Hein et al. (2009) and Darvill et al. (2015). We sampled quartzite cobbles for [10]Be analysis from the surfaces

5 of outwash grading to the Deseado 1, Deseado 2 and Telken 5 moraines. Similar samples were taken for Moreno 1 and 3 moraines and are presented in Hein et al. (2009). Using outwash cobbles has been shown to overcome issues of post-depositional erosion that affects moraine boulders in this area (Hein et al., 2009), but relies on finding locations where there is a clear geomorphic relationship between the glacial outwash and moraines in question. At suitable locations, cobbles were targeted based on quartz composition, size (6-15

10 cm and 300-1,000 g to contain sufficient [10]Be), and preservation (sub-rounded shape, little aeolian erosion, partially buried suggesting they were *in situ*). Surface lowering or up-freezing of cobbles can result in vertical movement through the outwash unit. In this region, it is more likely that cobbles will underestimate the true age of the unit (Hein et al., 2009; 2017). To circumvent this issue three or four cobbles were analysed at each sampling location, and the oldest taken as the best estimate of moraine deposition age.

As well as cobbles from the outwash surfaces, [10]Be depth profiles through exposures within the outwash units were used to estimate surface erosion rates and nuclide inheritance. We sampled through two outwash profiles relating to the Deseado 2 and Telken 5 moraines. Each of the samples making up the depth profiles consisted of 50 to 100 quartz-rich gravel clasts taken from horizontal lines, 2-4 cm wide at various depths below the surface. Five depths were sampled per profile between 50 cm and 240 cm deep; sufficient depth

to account for nuclide attenuation from the surface and reliably estimate any cosmogenic [10]Be inheritance. The top 50 cm of the profiles were carbonated due to post depositional processes but the rest of the profiles were of relatively consistent density.

### 2.2.2 Analytical methods

Surface cobbles were analysed individually as independent estimates of exposure age. The gravels at each

25 depth in the profiles were amalgamated to produce an average nuclide concentration for that depth. Samples were crushed and sieved to obtain the 125-250 $\mu$m and the 250-500 $\mu$m fractions, which were then purified using a Frantz magnetic separator to isolate non magnetic fractions. Following dissolution in concentrated





HF, the residual sample material was leached with 10mL $H_2O$ to extract $Be(OH)_2$ (Stone, 1998). A combination of anion and cation resins and precipitation was used to purify Be (Von Blanckenburg et al., 2004; Norton et al., 2008). The oxidized BeO was mixed with Nb powder and pressed into cathodes and measured on the 0.5 MeV Tandy accelerator at ETH Zürich (Müller et al., 2010). The resulting $^{10}Be/^9Be$ ratios were

normalized to S2007N ($^{10}Be/^9Be = 28.1 \times 10-12$; (Christl et al., 2013; Kubik & Christl, 2010)), with a $^{10}Be$ half-life of $1.387 \pm 0.012$ Ma (Chmeleff et al., 2010; Korschinek et al., 2010). Measured $^{10}Be/^9Be$ ratios were 30 - 500 times higher than the procedure blank ($9.60 \times 10^{-15} \pm 4.77 \times 10^{-15}$). Blank corrected $^{10}Be$ concentrations decrease exponentially with depth from greater than $36 \times 10^5$ for near surface samples to less than $2 \times 10^5$ at  2 metres depth.

### 2.2.3   Deposition ages calculations

Exposure ages for the outwash surface cobbles were calculated from $^{10}Be$ concentrations using version 2.3 of the CRONUS online calculator (Balco et al., 2008). Shielding was measured in the field (>0.999999 for every sample), a density of 2.65 g cm-3 was used for every sample, and no erosion rate was applied since there were no visible signs of surface erosion. We used the global $^{10}Be$ production rate ofBorchers et al.

(2016) and the 'Lm' scaling scheme (as described in Balco et al. (2008), Lal (1991) and Stone (2000) with paleomagnetic corrections from Nishiizumi et al. (1989)). However the scatter between the different scaling schemes is comparable to the external uncertainties.

We modeled the most probable deposition age, surface erosion rate, and nuclide inheritance for each depth profile using the Monte Carlo simulation approach of Hidy et al. (2010). A production rate of 6.2

atoms/g/yr was used, taken from an average of the scaling schemes in the CRONUS calculator. Changing this production rate by $\pm 0.3$ atoms/g/yr does not significantly alter the results. We used an average profile density of 2.5 g cm-3 (based on field observations), and a shielding factor of 1. The *a priori* values assigned to the Bayesian Monte Carlo simulation were: 0 to 1 cm/ka erosion rate with a maximum erosion threshold of 1000 cm and uniform distribution (based on erosion rates in Hein et al. (2009) for the Hatcher profile in

nearby Lago Pueyrredon and Darvill et al. (2015)); 0 to $10^5$ atoms/g inheritance with uniform distribution (based on concentrations in the lower levels of the profiles); 200 to 800 ka age range for Deseado 2 and 700 to 1200 ka age range for Telken 5, with uniform distribution. These ages were chosen based on the argon ages of Singer et al. (2004) and the Moreno age refinement of Hein et al. (2017).

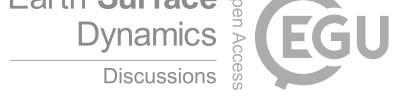

## 2.3 U-Th methodology

### 2.3.1 Principles and theory

The comminution age theory introduced by DePaolo et al. (2006) is a promising approach to estimating the age of sediment production. $^{238}$U decays into $^{234}Th$ with a half life of 4.5 Ga. During this $\alpha$ emission, the daughter isotope recoils over a distance of several tens of nanometers depending on the type of mineral (e.g. 30 nm for a feldspar). $^{234}Th$ quasi instantaneously produces $^{234}$U, with a longer half life of 245 ka: long enough to be measurable and usable on geologic timescales. If the $^{238}$U decay occurs on the edge of a sediment grain, the daughter isotope is ejected from the grain and a disequilibrium in the decay chain starts as the $^{234}$U/$^{238}$U ratio of the grain starts to decrease. If the grain is small enough, with a large surface area compared to its volume, the disequilibrium can be measured. Steady state is reached after a time characterized by the half life of $^{234}$U. During this transient state, the magnitude of the disequilibrium in a grain equates to the time since it was produced. The steady state disequilibrium value depends on the fraction of $^{234}$U ejected out of the grain due to $\alpha$ recoil. The process can be modeled using the following two equations describing the evolution of $^{238}$U and $^{234}$U:

$$\frac{dN_{238}}{dt} = -\lambda_{238}N_{238}$$

$$\frac{dN_{234}}{dt} = -\lambda_{234}N_{234} + (1 - f_\alpha^{234})\lambda_{238}N_{238}$$

$\lambda_{234}$ and $\lambda_{238}$ are the decay constants of $^{234}$U and $^{238}$U respectively. $N_{234}$ and $N_{238}$ are the number of nuclides for each isotope, and $f_\alpha^{234}$ is the recoil loss fraction of $^{234}$U (the proportion of $^{234}$U in the grain ejected during $\alpha$ decays). We discuss the estimation of this parameter in more detail in section 2.3.4. The evolution of the U/Th ratio over time is shown in Figure 3a (blue curves). The steady state disequilibrium model is based on the assumption that the sediment grain has not been weathered over time, or at least that the loss of nuclide due to weathering was negligible compared to loss by $\alpha$-recoil. The assumption may be valid for very arid areas, but is otherwise unlikely to hold true. However, the influence of weathering on the comminution age model has not been investigated.

In order to decouple the respective influence of $\alpha$-recoil and weathering, we measured $^{230}$Th concentration. The solubility of thorium is much lower than uranium. Therefore the ratio of $^{230}$Th/$^{234}$U can inform weathering rates. $^{230}$Th is produced by $^{234}$U decay with a half life of 75 ka. It can also be ejected out of



the grain due to $\alpha$-recoil. The recoil loss fraction of $^{230}$Th, $f_\alpha^{230}$ is related to $f_\alpha^{234}$ (Hashimoto et al., 1985; Neymark, 2011)

$$f_\alpha^{230} = f_\alpha^{234} \times \frac{\alpha_{230} M_{234}}{\alpha_{234} M_{230}} = 1.176 \times f_\alpha^{234} \tag{1}$$

$\alpha_{230}$ and $\alpha_{234}$ are the decay energy of $^{238}$U to $^{234}$U and of $^{234}$U to $^{230}$Th, respectively (71.79 and 82.96

5  keV respectively). $M_{234}$ and $M_{230}$ are the atomic mass of $^{234}$U and $^{230}$Th. This gives $f_\alpha^{230} = 1.176 \times f_\alpha^{234}$. The evolution of the number of nuclides $^{230}$Th is then given by

$$\frac{dN_{230}}{dt} = -\lambda_{230} N_{230} + (1 - f_\alpha^{230})\lambda_{234} N_{234} \tag{2}$$

Since $f_\alpha^{230} > f_\alpha^{234}$ and $\lambda_{230} > \lambda_{234}$, $(^{230}\text{Th}/^{238}\text{U}) < (^{230}\text{Th}/^{234}\text{U}) < (^{234}\text{U}/^{238}\text{U})$ is always verified. However, weathering would increase the ratio of $^{230}$Th/$^{234}$U and $^{230}$Th/$^{238}$U because Th is less soluble than U,

and likely decrease the ratio of $^{234}$U/$^{238}$U because radioactive decay places $^{234}$U is on a more fragile site compared to $^{238}$U so that it could be more easily weathered (DePaolo et al., 2012; Handley et al., 2013a). Consequently, Th isotopes can help to identify and quantify the effects of $\alpha$-recoil and weathering, respectively. New equations can be derived for the evolution of the different nuclide contents following Chabaux et al. (2003, 2008), and Dosseto et al. (2008) :

$$\frac{dN_{238}}{dt} = -k_{238} N_{238} - \lambda_{238} N_{238}$$
$$\frac{dN_{234}}{dt} = -k_{234} N_{234} - \lambda_{234} N_{234} + (1 - f_\alpha^{234})\lambda_{238} N_{238}$$
$$\frac{dN_{230}}{dt} = -k_{230} N_{230} - \lambda_{230} N_{230} + (1 - f_\alpha^{230})\lambda_{234} N_{234}$$

where $k_{238}$, $k_{234}$ and $k_{230}$ are the leaching coefficient of $^{238}$U, $^{234}$U and $^{230}$Th respectively. The solution

is shown in figure 3.

The steady state activity ratios can also be derived:

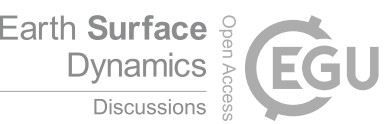

$$\left(^{234}U/^{238}U\right)_{steady} = \frac{\lambda_{234}\left(1 - f_{\alpha}^{234}\right)}{\lambda_{234} + k_{234} - k_{238}}$$

$$\left(^{230}Th/^{234}U\right)_{steady} = \frac{\lambda_{230}\left(1 - f_{\alpha}^{230}\right)}{\lambda_{230} + k_{230} - k_{238}}$$

$$\left(^{230}Th/^{238}U\right)_{steady} = \frac{\lambda_{230}\lambda_{234}\left(1 - f_{\alpha}^{230}\right)\left(1 - f_{\alpha}^{234}\right)}{\left(\lambda_{230} + k_{230} - k_{238}\right)\left(\lambda_{234} + k_{234} - k_{238}\right)}$$

Here, parentheses are used to represent activity ratios, corresponding to the isotopic ratio multiplied by the ratio of the respective decay constants of each isotope.

### 2.3.2 Sample preparation

Silt samples were collected from target moraines. Water has likely percolated through these silts since
moraine deposition, triggering the precipitation of secondary carbonates, oxides, organic matter and clays. These secondary fractions have different U and Th disequilibria than the primary silicate and it is therefore important to remove them before analysis without altering the primary silicate fraction. We used the protocol established by Lee (2009), with slight modifications to help overcome the low solubility of Th that can make it difficult to remove during successive leaching, and allow Th to adsorb onto mineral surfaces after ejection
from the grain. Our protocol is as follows:

1) The samples were sieved at 50 $\mu$m in order to collect the $\leq$ 50 $\mu$m fraction.

2) Approximately 10 g of this fraction was then weighed in a quartz crucible and heated for 4 hours at 550°C to burn organic matter. Lee (2009)'s optimal protocol uses $H_2O_2$ to dissolve organic matter, but other studies have shown that this reagent is not always fully selective (Tessier et al., 1979). The ashing of organic
matter yields similar results, but is normally conducted at the end of the procedure (Lee, 2009). By ashing first, we were able to process larger amounts of silt, limiting furnace contamination and aiding the process of dissolving of precipitates (mainly oxides formed during heating) in the following steps. Oxides turn the samples a brown to reddish color.

3) Sample grains were dispersed in 0.1 N sodium oxalate and the $\leq$ 4 $\mu$m fraction was removed using
Stokes settling. The $\leq$ 4 $\mu$m fraction consists of clays and also primary silicates with larger U-Th disequilibria.



4) For each sample, 4 g of sediment was weighed in a 50 mL centrifuge tube and leached twice with 32 mL 1M magnesium nitrate to remove the exchangeable fraction and any residues of the ashing process.

5) 32 mL of acetic acid buffered to pH5 with 1M sodium acetate was then added and agitated at room temperature during 5 hours to dissolve the carbonates.

6) 40 mL of 0.04M hydroxilamine hydrochlorid in 25% (v/v) acetic acid was added and heated at 95°C for 6 hours twice over to dissolve oxides. After this procedure, the samples still had a brown to reddish color from oxides formed in the furnace. These amorphous oxides can be resistant to the hydroxylamine hydrochloride treatment, as shown by Gontier (2014). These authors recommended using oxalic acid in ammonium oxalate (agitated in the dark at room temperature for 4 hours) (Leleyter & Probst, 1999). This procedure preserves silicate from dissolution (Gontier, 2014). Note that for the last three steps, the leaching residues were centrifuged and rinsed twice with mQ water.

At this stage the samples should contain only primary silicates between 4 and 50 $\mu$m. To evaluate whether sample preparation fully removed secondary phases (e.g. carbonates, oxides, clays) without affecting primary silicates, we made Scanning Electron Microscopy (SEM) images at the University of Strasbourg, shown in section 3.2. Our observations showed the method to be effective (see the Supplementary Information for further discussion on the efficacy of Th removal in this protocol) and the samples were deemed ready for disequilibrium analysis.

Two duplicates of this pretreatment protocol were measured (see Table 3). They are remarkably consistent within errors for ($^{230}$Th/$^{238}$U) and ($^{230}$Th/$^{234}$U) and only slightly different for ($^{234}$U/$^{238}$U) (only twice the external uncertainty). Since the surface area data are also slightly different, this small difference could be associated with heterogeneities in the powder, or fractionation of the grain size during the sampling of the powder, or biases in the leaching process.

### 2.3.3 U-Th disequilibrium analysis

A first aliquot of the samples was dedicated to U-Th analysis. This aliquot was powdered in an agate ball mill to optimize sample homogeneity and acid dissolution. Approximately 100 mg of this powder was spiked with $^{233}U$-$^{229}Th$ tracer and digested with concentrated HF-HNO$_3$-HClO$_4$ acids following Pelt et al. (2013). Separation and purification of U and Th followed standard anionic resin chromatography (Dequincey et al., 2002; Granet et al., 2007; Pelt et al., 2008). U and Th isotopic ratios and concentrations were determined





by standard-sample bracketing (SSB) using an MC-ICPMS Neptune, following an optimised procedure upgraded from previously published protocols (Pelt et al., 2008; Ma et al., 2012; Bosia et al., 2016). The $^{233}$U-$^{229}$Th spike was calibrated against the gravimetric NIST SRM 3164 and 3159 U and Th pure solutions. IRMM-184 and IRMM-035 isotopic standard solutions were spiked and used as bracketing solutions

for U and Th SSB analysis, with the $^{233}$U/$^{234}$U and $^{229}$Th/$^{230}$Th ratios of these mixtures calibrated by TIMS Triton. We used the consensus value from Sims et al. (2008) for the $^{232}$Th/$^{230}$Th ratio of the IRMM-035 standard and our own absolute TIMS Triton measurements for the $^{234}$U/$^{238}$U ratio of the IRMM-184 standard instead of the certified values. The peak-tailing of the $^{232}$Th over the $^{230}$Th was corrected using an exponential law fitted by signals measured in 229.6 and 230.6 masses. The precision and accuracy of the data

were estimated from regular analysis of international pure solutions (HU1 for ($^{234}$U/$^{238}$U): 1.0008±0.0005 ($2\sigma$, N=5), IRMM-036 for $^{232}$Th/$^{230}$Th: 329064±1313 ($2\sigma$, N=5)) and rock basalts (BCR-2, AThO and BE-N). . Results are consistent with published data and the $^{238}$U-$^{234}$U-$^{230}$Th secular equilibrium assumed for old (several Ma) BCR-2 and BE-N 'unweathered' basalts, within errors. Based on these pure solutions and homogeneous basalt powders we estimate an uncertainty (2SD) of 0.2% for ($^{234}$U/$^{238}$U), 0.5-1% for

($^{230}$Th/$^{232}$Th) and  1-1.5% for ($^{238}$U/$^{232}$Th), ($^{230}$Th/$^{238}$U) and ($^{230}$Th/$^{234}$U). A duplicate of the protocol was measured and agree within the internal measurement uncertainties for ($^{234}$U/$^{238}$U), ($^{230}$Th/$^{238}$U), ($^{230}$Th/$^{234}$U) (see Table3). Usual blanks in the lab are in the range of 20-50 pg for U and 100-200 pg for Th and therefore negligible here.

### 2.3.4   Estimation of the recoil loss factor

The recoil loss parameter ($f_\alpha$) is a critical parameter in comminution age theory. Different methods have been proposed to estimate this parameter, summarized in Maher et al. (2006) and Lee et al. (2010), but discrepancies up to an order of magnitude can be found between the different methods(Handley et al., 2013a, b). Here we used the $N_2$ gas absorption technique, which consists of surface area and fractal dimension measurements. Other methods are more subjective in that they involve visual estimation and assumptions for

the aspect ratio and surface roughness coefficient. We measured the specific surface area for all samples and assessed the efficiency of this method to determine $f_\alpha$ and the relation between $f_\alpha$ and U-Th disequilibria.

A second aliquot of the samples (at least 1 g per sample) was prepared for these analyses, conducted at the Institut de Physique du Globe de Strasbourg on a Carlo Erba Sorptomatic 1990 machine. Details of





the procedure are given in the Supplementary Information. Fractal dimension were also measured for each sample and the recoil loss factor $f_\alpha$ calculated following Bourdon et al. (2009), after Semkow (1991):

$$f_\alpha = \frac{1}{4}\left[\frac{2^{D-1}}{4-D}\left[\frac{a}{R}\right]^{D-2}\right] R.S.\rho_s \tag{3}$$

Where $S$ is the specific surface area, $D$ is the fractal dimension, a is the diameter of the adsorbate molecule
(0.35 nm), $R$ is the recoil lenght (30 nm), and $\rho_s$ is the density of the solid (2650 kg.m$^{-3}$).

### 2.3.5   Monte Carlo analysis

A Monte Carlo approach was used to determine weathering intensities during both the pre- and post-deposition histories of the different moraine silts, and the duration of this pre-deposition history (called 'recycling time' from this point forward). We used the theoretical scheme described in section 2.3.1 to es-
timate these weathering intensities and recycling time. Known or informed parameters in the Monte Carlo simulations were the activity ratios ($^{234}$U/$^{238}$U) and ($^{230}$Th/$^{238}$U), initial activity ratio (from the glacial flour sample), recoil loss factor $f_\alpha$ and deposition ages of the moraines (the post-depositional duration). Unknown parameters were the weathering intensities of $^{238}$U, $^{234}$U and $^{230}$Th for the pre- and the post-deposition history. The relative weathering intensity of $^{234}$U compared to $^{238}$U, and $^{230}$Th compared to $^{238}$U, were
assumed to be the same before and after moraine deposition. Consequently, we needed to determine five unknowns: the respective weathering coefficients $k_{238}^{pre}$, $k_{238}^{post}$, $k_{234}/k_{238}$, $k_{230}/k_{238}$ and the recycling time $t_{recycl}$ using Monte Carlo simulations. We chose a large, arbitrary number of values for the unknowns and calculated the activity ratios associated with these set of values. Then, we calculated a misfit $M$ for the data that is the difference between measured and modeled values, weighted by the uncertainty. The lower the
misfit, the larger the probability of this set of values (we show the likelihood L for better visibility):

$\quad$ M=(m-o)$^T \sigma^{-1}(m-o)$

$L = exp(-M/2)$ Where $m$ is the vector of modeled parameters (the five unknowns above), $o$ is the observed values (the activity ratios ($^{234}$U/$^{238}$U) and ($^{230}$Th/$^{238}$U)), and $\sigma$ is the diagonal matrix of uncertainties on the data. In theory, an optimal solution can be determined, corresponding to the lower misfit (larger likelihood).
However, no solutions were found because the number of unknowns was too great. We also know the relative weathering intensities of $^{238}$U, $^{234}$U and $^{230}$Th : since Th is less soluble than U and $^{234}$U is on more fragile



mineralogic sites. Hence, $k_{230}$ is lower than $k_{238}$, and $k_{234}$ is slightly larger than $k_{238}$. This information was used to further constrain the Monte Carlo simulations. For different couples of $k_{234}/k_{238}$ and $k_{230}/k_{238}$, we ran a Monte Carlo optimization to find an optimal solution (lower misfit). All optimal solutions for $k_{238}^{pre}$, $k_{238}^{post}$ and the recycling time are shown in Figure 6, and we use the optimal solutions to characterize the

pre-depositional duration of these sediments.

To test whether exchangeable Th might not be efficiently leached by our chemical protocol, we also made a simulation with $f_\alpha^{230} = 0$ and $f_\alpha^{230} = \dfrac{1.176 \times f_\alpha^{234}}{2}$ (half of the theoretical value described in section 2.3.1). If Th was totally readsorbed onto the mineral surfaces after ejection from the grain (and not subsequently removed during leaching), then it is equivalent to having no ejection ( $f_\alpha^{230} = 0$). The case with $f_\alpha^{230} =$

$\dfrac{1.176 \times f_\alpha^{234}}{2}$ is an intermediary between total removal of the adsorbed Th and no removal at all, discussed further in the Supplementary Information.

## 3  Results

### 3.1  Cosmogenic ${}^{10}$Be and deposition ages

The results of ${}^{10}$Be analyses are shown in Figure 2 and Tables 2 and 1. Refer to supplementary information

for more details on the profiles ages results.

Eight surface cobbles from outwash relating to the Moreno 1 and 3 moraines yielded ${}^{10}$Be exposure ages ranging from 168-269 ka (published in Hein et al. (2017)). To these we add six more surface exposure ages and two depth profile ages from outwash related to the Deseado and Telken moraines. Three cobbles from Deseado 1 produced relatively tightly-clustered exposure ages of 430-468 ka. A cobble from the ice-distal

Deseado 2 outwash yielded a much younger, stratigraphically-inconsistent age of 293 ka (sample D2-T12), similar to the Moreno ages. Two further cobbles from the Deseado 2 outwash yielded ages of 520 ka and 618 ka, the oldest of which agrees, within errors, with a modelled depth profile age of 600 ka (+70/-35) from the same outwash unit. The agreement between older ages from surface cobbles and the Deseado 2 depth profile is unsurprising given that modelling suggests surface erosion ($<0.2$ cm.ka$^{-1}$) and inheritance within

the outwash sediments ($<2.10^4$ atomg.g$^{-1}$) was relatively low (see Supplementary Information). There are a number of factors that can result in some age scatter within outwash surface cobbles, including surface deflation and up-freezing of clasts (Darvill et al. (2015); Hein et al. (2017)). Moreover, Hein et al. (2009,



2017) suggested that the oldest surface exposure ages from outwash in this region will most closely relate to deposition age. The D2-T12 sample is >200 ka younger than the other Deseado 2 cobbles, and it is likely that this cobble was either deposited at a much later time (perhaps during Moreno-stage glaciation, although the difference in altitude makes this unlikely) or was not in situ. The close agreement between other Deseado 2 ages supports the removal of D2-T12 as an outlier. Finally, a modelled depth profile through outwash relating to the Telken 5 moraine produced an age of 780 ka (+170/-80, $1\sigma$ confidence interval), with relatively low surface erosion (<0.1 cm ka-1) and inheritance (<$6.10^4$ atomg.$g^{-1}$). This age is within the lower range of 760-1016 ka known from radiometric ages (Singer et al., 2004). Sensitivity tests showed that modifications of the a priori values of erosion rates and inheritance lead to insignificant changes in the resulting age.

## 3.2 Evaluation of the U-Th chemical procedure with SEM images

For three samples taken randomly, we looked at SEM images after $50\mu m$ sieving and preparation (Figure 4) to assess the effects of our chemical and mechanical treatment. We were particularly interested in whether secondary phases could remain after processing, and if the primary silicates could have been altered by this treatment. The minerals observed are mainly quartz, feldspars and micas, corresponding to the lithology of the Andes in these latitudes. We list below the main observations :

- No trace of carbonate, organic matter or oxides were observed in samples that underwent the entire protocol. Prior to treatment, we only note the presence of oxides (Figure 4b). This does not, however, preclude the presence of carbonate or organic matter since a difference of mass before and after steps 2 and 5 of the protocol were observed. Therefore, the protocol seems efficient at eliminating carbonates, oxides, and organic matter.

- Samples were generally cleaner after preparation. This is because $\leq 4\ \mu m$ fractions, containing mainly clays, were removed by Stokes settling. However, clays are still observed in the samples after preparation (see Figure 4b and c), mainly agglomerated around primary grains as clay pellets. In one sample, up to one third of the grains have clay pellets. These clays could have been precipitated directly at the surface of the grains during weathering or agglomerated subsequently. It is difficult to fully eliminate the clays and a potential bias could be introduced if too many clays remain in the sample. This po-





tentially limits the application of comminution ages to samples with very low amount of clays, since
their complete removal is not currently possible.

– Most micas show surfaces having experienced weathering (Figure 4d). The shapes observed are typical
of weathering both before and after the protocol. This shows that our silts samples have experienced
weathering and that it could potentially have affected U-Th disequilibrium.

– Following the preparation protocol, two feldspar grains in one sample displayed evidence of corrosion
(Figure 4e and 4f). While this damage could be associated with alteration of the primary silicates
during leaching, these are the only two corroded grains visible in the samples we tested. Since the
corrosion pits are smaller than $4\mu$m, and all the grains are larger than $4\mu$m in our samples, ubiquitous
damage would be observable on other grains. As such, we assume that primary mineral corrosion
during the protocol is minimal.

### 3.3 U-Th - $f_\alpha$ - age: a 3 dimensional problem

U-Th results are shown on Figure 5 and Table 3. ($^{234}$U/$^{238}$U) from moraines display little variation, ranging
from 0.95 to 0.97, while glacial flour has a composition of 1.007, close to equilibrium. For the moraine sam-
ples, a general decrease of ($^{234}$U/$^{238}$U) is observed with time, though with significant noise. ($^{230}$Th/$^{238}$U)
vary between 0.94 and 0.97 with he youngest moraine sample having a composition around 1. Similarly
($^{230}$Th/$^{234}$U) mostly range from 0.99 and 1.01, with the youngest moraine around 1.03. Again, an over-
all decrease in ($^{230}$Th/$^{234}$U) is observed with time. The glacial flour is close to equilibrium compositions
(($^{230}$Th/$^{238}$U)=1.02 and ($^{230}$Th/$^{234}$U)=1.01). Specific surface areas vary from 1.5 to 7 $m^2/g$, which is typ-
ical for this kind of samples, and the fractal dimensions range from 2.45 to 2.6. The recoil loss factors $f_\alpha$
calculated using equations 3 are between 0.005 and 0.025, with no observable trend with time.

Handley et al. (2013a) analyzed only a few sample with the gas adsorption technique to characterize the
specific surface area and $f_\alpha$. We analyzed every sample with this technique, highlighting the consistency
of the noise in ($^{234}$U/$^{238}$U) and the dispersion in $f_\alpha$ derived from specific surface area (Figure 5). The
relationship between ($^{234}$U/$^{238}$U), $f_\alpha$ and age is described by a surface in three dimensions. In other words,
the noise observed in the ($^{234}$U/$^{238}$U)-age diagram, around the general decreasing pattern is in a large part
associated with the dispersion of $f_\alpha$. This suggests that specific surface area data are consistent and reliable,




or that, if a bias exist in these data, it is systematic. Such a systematic bias would arise from a rather random event due to the sample processing, so this latter explanation seems unreasonable. We conclude that the specific surface area determination based on the gas adsorption technique is valid to evaluate the recoil loss factor in the U-Th comminution age theory.

We observe that $(^{230}\text{Th}/^{234}\text{U})$ is larger than $(^{234}\text{U}/^{238}\text{U})$ and $(^{230}\text{Th}/^{238}\text{U})$ has comparable compositions to $(^{234}\text{U}/^{238}\text{U})$. As mentioned in section 2.3.1, this observation is incompatible with a simple comminution age model (only $\alpha$-recoil). An enrichment of $^{230}\text{Th}$ compared to $^{234}\text{U}$ could be the imprint of weathering on silts found in the moraines. Similarly, the incompatibilities of $(^{234}\text{U}/^{238}\text{U})$ and $f_\alpha$ in the framework of the simple comminution age model suggest that weathering must be considered. In the comminution age theory

as described by DePaolo et al. (2006), $1 - f_\alpha$ must always be lower than $(^{234}\text{U}/^{238}\text{U})$, and for a sample old enough to have reached steady state, $(^{234}\text{U}/^{238}\text{U})=1 - f_\alpha$. This is not what we observe, similarly to Handley et al. (2013a). However, as described in section 2.3.1 such an observation can be explained if weathering is considered and $k_{234} \geq k_{238}$.

    Given the large disequilibrium of the youngest samples in $(^{234}\text{U}/^{238}\text{U})$ and in $(^{230}\text{Th}/^{234}\text{U})$ (Table 3 and

15 Figure 5), and the fact that the glacial flour (the most probable inital composition for the moraine silts) is close to equilibrium, we cannot ignore the pre-depositional history (i.e. before deposition in the moraines). Based on a starting value close to equilibrium, the disequilibrium values measured here cannot be reached in only 20, or even 200 ka. Since our estimations of the $f_\alpha$ seem consistent and reliable, this requires consideration of a pre-depositional history involving weathering with a different intensity than during the

20 post-deposition history.

## 3.4   Predeposition history and Monte Carlo analysis

We attempt to constrain the pre-depositional history using a Monte Carlo analysis. The results show an inverse relationship, with the greatest probability for $k_{238}^{pre}/k_{238}^{post}$ between 2 and 3 and a recycling time between 100 and 200 kyrs. The optimal values of $k_{234}/k_{238}$ and $k_{230}/k_{238}$ associated with this solution describe

an anti correlation with $1.2 \leq k_{234}/k_{238} \leq 2$ and $0.01 \leq k_{230}/k_{238} \leq 0.8$ (figure 6). These parameters are, however, poorly constrained by the inversion process.

    Despite the numerous parameters to be constrained, the model converges for key results as shown above. We show in particular that, in the framework of the 'modified comminution age theory', incorporating weath-



ering, described in section 2.3.1, we can say that the silts from the LBA moraines were likely eroded 100 to 200 kyrs before being deposited in the moraines and that, on average, they experienced around 2-3 times more intense weathering during this interval than after deposition (Figure 6). The best fit, corresponding to a recycling time of 180 kyrs, a ratio $k_{238}^{pre}/k_{238}^{post} = 2.35$, $k_{234}/k_{238} = 1.4$, and $k_{230}/k_{238} = 0.6$, is shown on supplementary figure 5.

## 4 Discussion

### 4.1 Glacial chronology and perspectives on the control of ice extent in the Patagonian Andes

The new 10Be exposure ages in this study help to clarify the timing of deposition of several of the LBA moraines. The oldest outwash cobbles are taken as closest to the age of deposition because depth profiles show that average nuclide inheritance is relatively low and outwash surfaces show evidence for deflation (exposing younger cobbles; Hein et al. (2011, 2017)). It is possible that even the oldest cobbles underestimate the age of deposition if they have also been exhumed, and we apply no erosion correction to our exposure ages. As discussed in Hein et al. (2017), outwash surface cobbles imply that the Moreno moraines were deposited at ca. 260-270 ka, during Marine Isotope Stage 8.

Unlike the Moreno system, surface cobbles from the Deseado moraines appear to date from two different glacial cycles. Exposure ages from Deseado 1 outwash yield relatively tightly clustered ages of 430-470 ka, suggesting that the limit was deposited during MIS 12. The older of two published moraine boulder exposure ages from the Deseado 1 moraine (erosion-corrected to 476 ka; Kaplan et al. (2005)) is consistent with the cobble ages in this study. In contrast, surface cobbles from Deseado 2 outwash yield ages of 520 ka and 618 ka (excluding the anomalously young D2-T12 age of 293 ka). The dating would be less conclusive without the accompanying depth profile age of 600 +70/-35 ka. Taken together, the oldest surface cobble and depth profile suggest that the Deseado 2 limit was deposited at ca. 600-620 ka. Within errors, the limit may relate to MIS 16. The scatter in ages is interesting given that luminescence dating yielded an even younger age of 123 ± 18 ka for the Deseado 2 limit Smedley et al. (2016). This scatter may imply greater sediment reworking or a more complex relationship between moraines and outwash in this sequence. The Telken 5 depth profile gave an age of 780 +170/-80 ka that is stratigraphically consistent within our chronological dataset, but less helpful in determining when the limit was deposited. The error range spans MIS 18-24 (and





radiometric datings gave a lower age of 760 ka for the Telken serie), with larger probability around MIS20, so it is possible that the moraine was deposited during MIS 20. Since the entire Telken moraine system must be older than 760 ka (Singer et al., 2004), this implies that Telken 1 to 4 are also that age, and that the Telken moraines represent different glacial advances during the same glacial cycle.

In summary, there appears to be a pattern in the timing of moraine deposition in which alternate glacial cycles are represented in our chronology: Telken 5 during MIS 20; Deseado 2 during MIS 16; Deseado 1 during MIS 12; and the Moreno moraines during MIS 8. Previous work has shown that the innermost Fenix moraines relate to the Last Glacial Maximum during MIS 2 (Kaplan et al. (2004); Douglass et al. (2006); Smedley et al. (2016)). We strongly caution that scatter in surface cobble ages and error ranges in

depth profiles may complicate this pattern, particularly for the older limits. Moreover, we did not analyse the enigmatic Deseado 3 moraine and so cannot say whether this relates to the counterparts dated here or intervening glacial stages. However, dating of the Moreno and Deseado 1 and 2 moraines does imply that the intervening glacial cycle is absent from the record.

    The pattern in timing of moraine deposition around Lago Buenos Aires implies that there are a few glacial

cycles that are either not recorded in this area, or have been erased or removed. Either the ice lobe did not advance during alternate glacial cycles, or advanced to similar or less extensive positions so that moraines and outwash were then removed by the following advance. A similar pattern was observed by Hein et al. (2009, 2011) for the Lago Pueyrredon ice lobe, 100 km to the south, and so there could be a regional driver of alternate glacial advances. One possibility is that erosion over successive glacial cycles caused entrenchment

of ice lobes within large basins on the eastern side of the Patagonian Ice Sheet (Kaplan et al., 2009). This erosion model has been linked to the pattern of nested limits seen across the former ice sheet (Kaplan et al., 2009; Anderson et al., 2012) but may be more complex if only alternate cycles are represented in the moraine record. Alternatively, a purely climatic forcing could have caused alternate strong advances, although there is no simple relationship between available climate records and alternate glacial advances at Lago Buenos

Aires. A complex erosion-climate feedback mechanism may determine when or how far glacial advances occurred in this region, but more detailed glacial models are required to further test such a model.

    The chronology presented here suggest that moraines reworking has probably occured in the area, and/or that glacial erosion in the Andes may feedback into the glacial lobe advance, to produce this observed pattern of absent intervening glacial cycles. Regardless, this chronology allows us to broadly constrain deposition



ages of moraines targeted for U-Th analysis, which may in turn inform sediment recycling times between preserved moraines.

## 4.2 Implications for sediment recycling, chemical weathering, climate and their interactions, in proglacial systems

Following the conclusions of Handley et al. (2013a), this residence time of 100-200 kyrs could be an artifact due to an addition of old dust. In our area, when a new moraine is being formed, the older sediments are on the east, whereas the dominant winds comes from the west. So it is unlikely that these winds could add older material to the one being deposited.

The long time estimated using the whole set of data and the Monte Carlo simulations, along with the
10 absence of inheritance in the [10]Be data, suggest that the sediment was not exposed at the surface in the proglacial system before being deposited in its moraine. This implies that the sediment must have been buried, either in the proglacial lake, or within a sedimentary pile (moraine, till, channel, etc. at least a few meters below the surface) before final deposition.

The dated sediment was most probably eroded during glacial periods, as erosion rates are much larger
than during interglacial. Likewise, deposition in an end moraine occurs during a glacial period. This means that the sediment deposited in the moraine may spend on average as long as 100-200 kyrs in the proglacial system (till, lakes sediments, etc.) before being deposited in the moraines. Hence the sediment eroded during a glacial period is, on average, deposited in the frontal moraine during the next glacial cycle. Using post-glacial sediment budget and reservoir theory, Hoffmann & Hillebrand (2016) modelled the residence time
of sediments in a periglacial system in the Canadian Rocky Mountains, and also found a time of 100 kyrs. This average residence time implies that a fraction of the sediment deposited in a moraine may have been produced during the same glacial cycle and with an other fraction being much older.

This time appears to be long. However, our measurements of cosmogenic exposure ages using [10]Be nuclide concentrations in outwash cobbles and profiles suggest that the preserved moraines represent every
25 other glacial cycle, indicating an advance over the proglacial sediment of one or two previous advances. Moreover, it has been shown that proglacial lakes occupying overdeepenings are filled following deglacia-tion (Eyles et al., 1991; Houbolt & Jonker, 1968). The lakes are filled with sediments during interglacial periods, and emptied during the subsequent glacial periods. These processes imply sediment erosion and



deposition over a full glacial-interglacial cycle which is consistent with both $^{10}$Be and U-Th disequilibrium data.

We also obtain weathering that is 2-3 times stronger during the 100-200 kyrs of the predepositional phase than after deposition. Based on the argument above, these sediments are exposed to water in the proglacial system before deposition in the moraine. Authors such as Anderson (2005) have shown that proglacial environments favor weathering. Since weathering rate is a time dependent parameter (e.g. White & Brantley (2003)), this recycling time of 100-200 kyrs has a likely non-negligible impact on weathering fluxes.

Understanding how sediments are evacuated and transported to the oceans, and when they experience weathering would help to quantify the relationships between erosion and climate. In particular, if 100 to 200 kyrs are needed to escape the periglacial area, important lags could be observed between the erosion forcing, and the weathering and climate response. This suggests that measured weathering variations could have occured over the last glacial-interglacial cycles, or have been smoothed/damped because of a lag in the sediment transport. Estimating these lag times would help understand the relationships between erosion, climate, and marine biogeochemical cycles. Erosion rates can be determined using thermochronology or cosmogenic isotopes. Sedimentation rates in the ocean can be estimated as well using marine core dating methods. The link between erosion and sedimentation is poorly known (Sadler, 1981) especially because the transport times are poorly constrained. This is exacerbated since the estimation of denudation rates using cosmogenic nuclides usually necessitates the assumption that the exposure to radiation after denudation is negligible. The transport time of 100 to 200 kyrs presented here suggests that the pathway between initial erosion and deposition is potentially complex. An effort to constrain these transport times appears to be potentially fruitful to reveal the actual link between erosion/sedimentation rates and climate.

## 4.3   Perspectives on comminution age method and its applications to characterize sediment transfer

Our data let us estimate only a mean recycling time for moraine sediment. Here we have discussed it in terms of recycling of the previous glacial cycle moraine, but it could also be interpreted as a mixing of much older, deeper, reworked sediment with new freshly eroded sediment. In other words, we do not quantify the amount of sediment escaping the proglacial system, or the amount of sediment which is deposited in the moraine rapidly after erosion. It would be beneficial to think about ages distributions, and not only mean





ages. However, this necessitates being able to measure U-Th disequilibrium on single grains, which remains an analytical challenge (Bosia et al., submitted).

Quantifying or reducing the effects of weathering remains a major challenge. Pure primary minerals with no clays would minimise these secondary processes. To this end, working on pure zircons grains could be an option. This would require improved mineral separation methods for fractions smaller than 50 $\mu$m. One would also have to assume that comminution effectively occured on such a heavy mineral, and that there is no initial disequilibrium (a problem being that zircons are much enriched in U than there surrounding minerals).

## 5  Conclusions

Our study is a first attept to quantify long term sediment transfer times in proglacial area. We take advantage of the particularly well preserved series of nested moraines of the Lago Buenos Aires in Patagnia. Our approach involves $^{10}$Be exposure dating of these moraines combined with U-Th disequilibrium measurement on the fine fraction of moraine sediment, within a 'modified' comminution theoretical framework, to characterize the pre-deposition and post-deposition histories of the sediments.

We show that weathering cannot be neglected when determining the comminution age. Measuring Th isotopes helps constrain the weathering process. Even with this complication, it remains possible to calculate comminution ages if the studied samples have well constrained deposition ages and have experienced the same predepositional history. In this case, the weathering intensities and the predepositional history duration are the same for all the samples. We also show that specific surface area measurements based on the gas adsorption technique for estimation of the recoil loss factor is a reliable method, and should be applied to all samples. One caveat is that clay minerals may not be removed from the samples, especially clay pellets agglomerated at some mineral surfaces, which could be a strong limitation of the use of this method to calculate transport times.

Our new chronology shows that some glacial cycles are either not recorded or not preserved in the LBA area. The associated moraines may have been removed by following glacial advances or may have been deposited farther upstream. Hein et al. (2011) found a similar pattern in Lago Pueyrredon area, suggesting

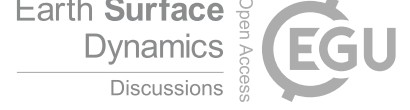

that the mechanisms driving glacial advance may have a regional extent. Future work may reveal wether erosion feedbacks in the Andes could be responsible for this pattern.

The absence of moraine records for a few glacial cycles in the LBA area can be explained by our U-Th data showing that there has been reworking/recycling of the sediments. Using a Monte Carlo approach we estimate that the silts from the Lago Buenos Aires frontal moraines system have 100 kyr residence time in the proglacial system (transported in the lake sediments, remobilized from the previous moraines or stored in intermediate reservoirs), and experienced three times more intense weathering before deposition than in the moraine. Our data represent a step toward the effort of constraining the pathways and timescales over which sediments are transported from source (erosion in mountain belts) to sink (ocean sediments). Although considered as a 'daunting challenge' by Sadler & Jerolmack (2015) and requiring additional development, this approach can contribute to our understanding of basin-scale sediment budgets and how erosion, weathering, and sedimentation evolve through time.

*Code availability.* TEXT

*Data availability.* TEXT

*Code and data availability.* TEXT

*Author contributions.*

*Competing interests.* The authors declare that they have no conflict of interest.

*Disclaimer.*





*Acknowledgements.* We thank the administration of the Laguna San Rafael National Park for delivering us the authorization of sampling in the glacier Los Exploradores area and especially Cristobal Fabres for his nice support and escort during the sampling.

Earth **Surface**
Dynamics
Discussions

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





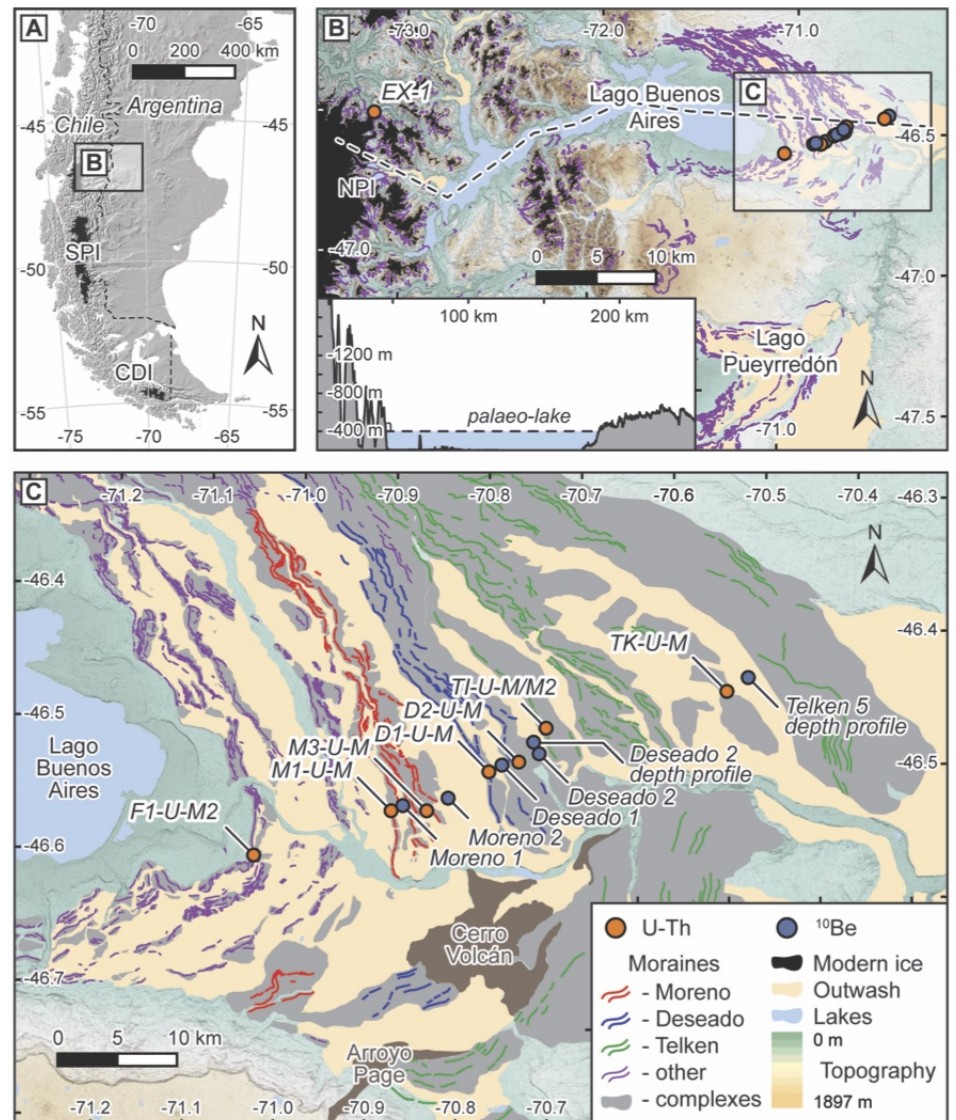

**Figure 1.** Map of the study area, adapted from Bendle (in press). [a] Southern South America showing the location of Lago Buenos Aires. [b] The series of frontal moraines sampled in this study. Silt samples are shown by red circles and cobbles and profiles for cosmogenic analysis are shown in blue; the glacial flour sample at the mouth of Los Exploradores Glacier is shown; and the extent of the present day ice field is shown in black. [c] West-east transect through the study area, from the Patagonian Andes to the frontal moraine systems of the proglacial lake (adapted from Kaplan et al. (2009)); sample locations as per [a]. Successive glaciations have formed the large over-deepening in which the proglacial lake has formed.



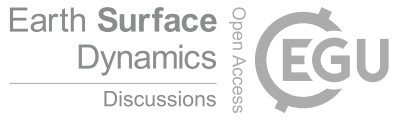

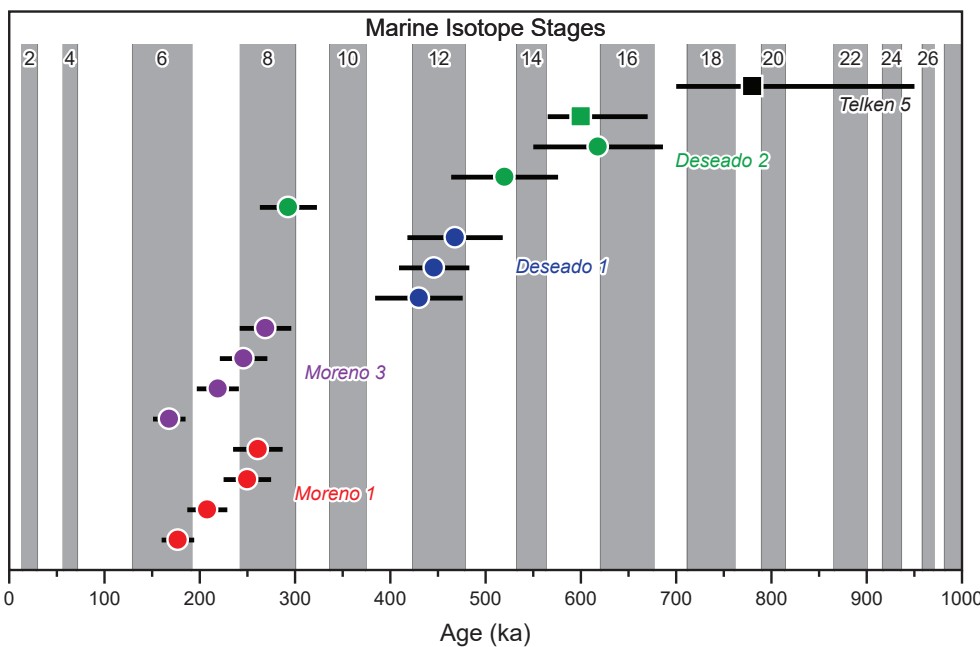

**Figure 2.** [10]Be exposure ages from outwash surface cobbles (circles) and modeled depth profiles (squares) for five of the Lago Buenos Aires moraines. Depth profile uncertainties are $1\sigma$. Grey shading shows glacial stages from the Marine Isotope Stage stratigraphy of Lisiecki  Raymo (2005).



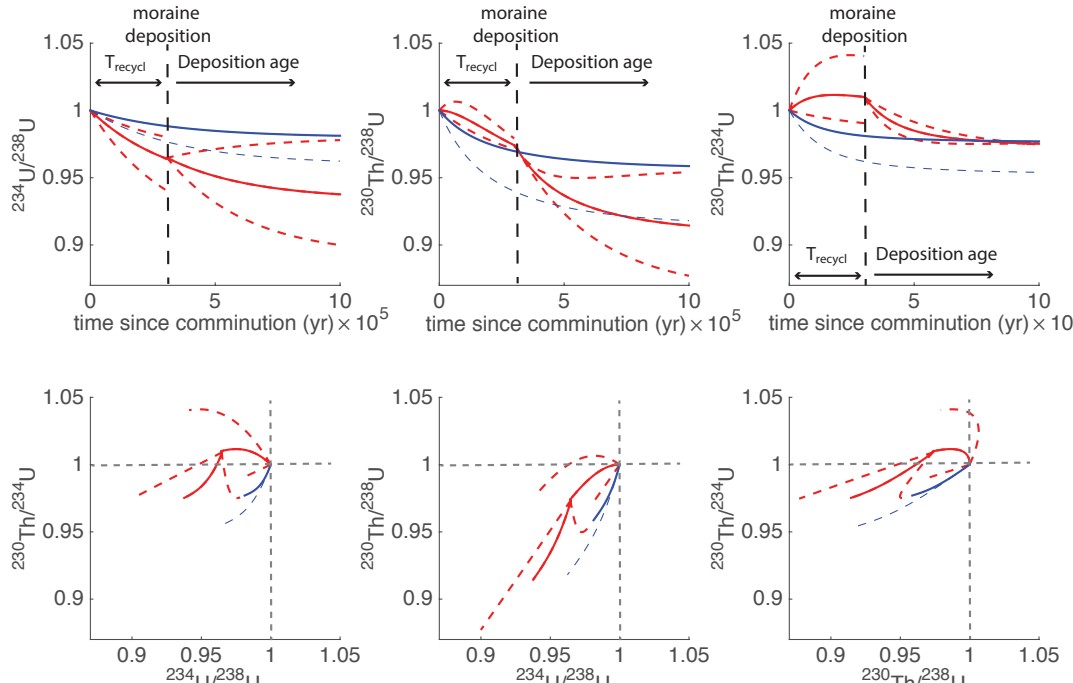

**Figure 3.** U-Th comminution results. (Upper panels) the time evolution of $^{234}$U/$^{238}$U, $^{230}$Th/$^{238}$U and $^{230}$Th/$^{234}$U activity ratios. (Lower panels) the co-evolution of these ratios in the framework of the comminution age model, with (red) and without weathering (blue). The blue lines represent the case as described by equations 2.3.1 to 2 (see DePaolo et al. (2006)), with only $\alpha$-recoil (solid lines: $f_\alpha = 0.02$ ; dashed lines: $f_\alpha = 0.04$). The red lines represent the case where weathering is considered, with two different weathering intensities (dashed and solid lines; always with $f_\alpha = 0.02$). The vertical dashed black lines show moraine deposition, the age of which is determined by the $^{10}$Be exposure ages. Weathering intensities are lower after moraine deposition. We assumed that $k_{234} \geq k_{238}$ and $k_{230} \leq k_{238}$, both before and after deposition. These activity ratios can follow complex paths, but the use of $^{230}$Th allow us to identify whether simple $\alpha$-recoil and/or weathering occurs.



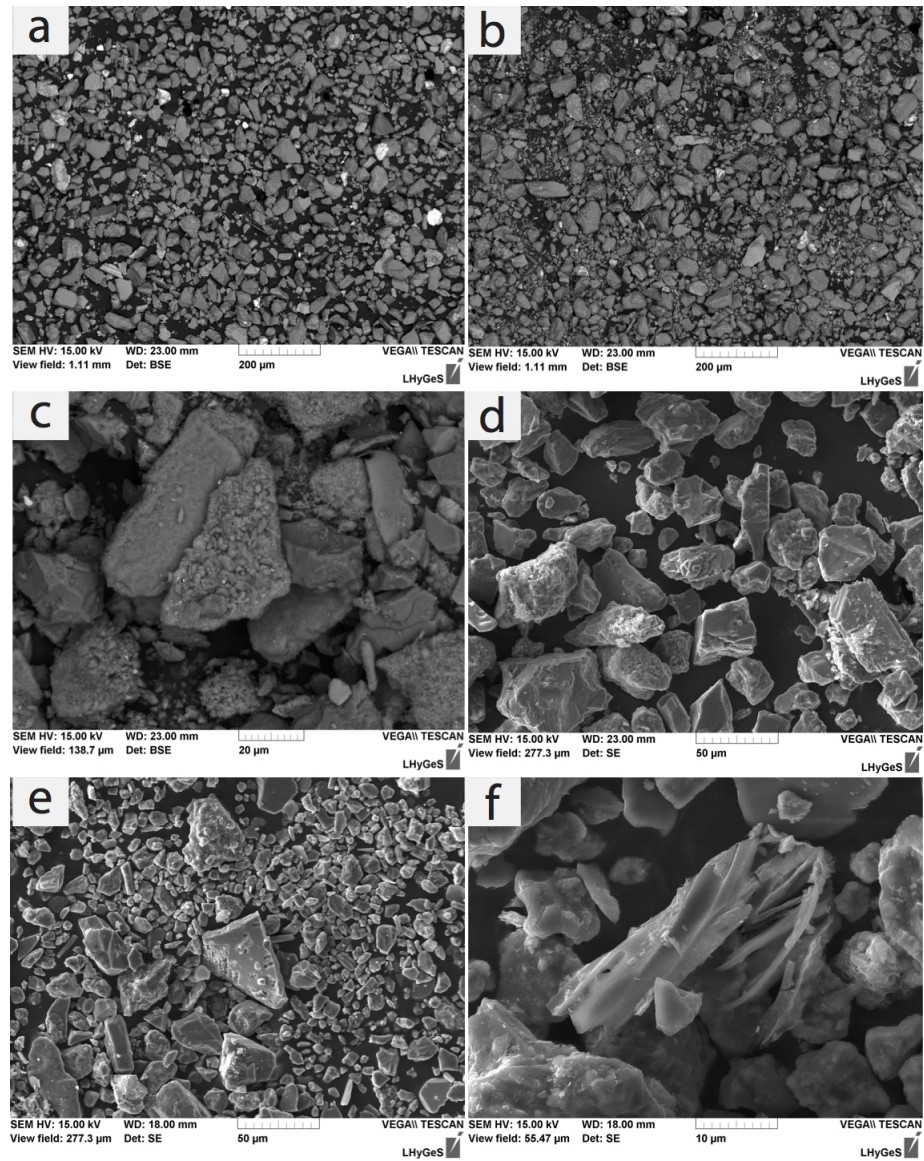

**Figure 4.** Scanning Electron Microscopy (SEM) images of selected samples. [a] Overview of a sample after the preparation process described in section 2.3.2. Grains vary in size and shape between 4 and $50\mu m$ and angular to subangular. White grains are heavy minerals such as zircon or oxides. [b] The same sample before $50\mu m$ sieving. The only notable difference is the presence of clays. [c] Zoomed image of a mineral grain covered in clays pellets that were not removed by the preparation process. [d] Mica grains with cauliflower faces, characteristic of weathering. [e] and [f] Feldspar grains with corroded surfaces. It is unclear whether these corrosion features result from geological processes or the sampling procedure, but they were only observed twice.



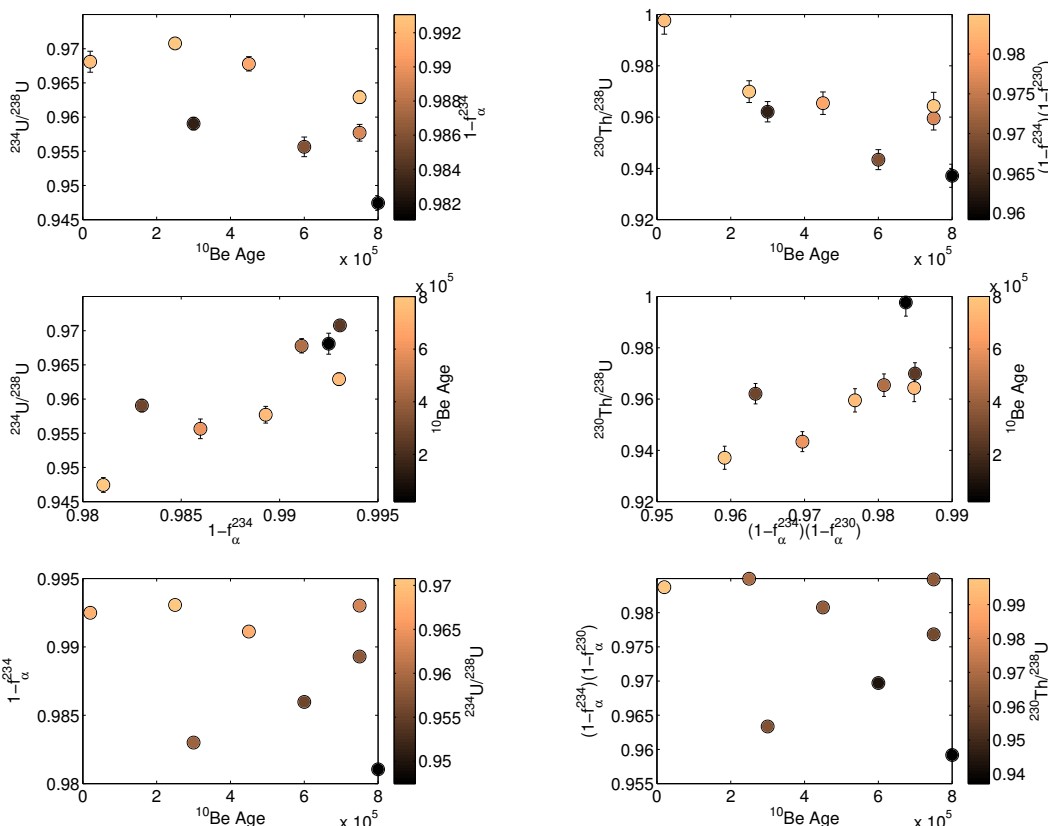

**Figure 5.** 2D cross plots of $^{234}$U/$^{238}$U and $^{230}$Th/$^{238}$U activity ratios; associated recoil loss factors $(1 - f_\alpha^{234})$ and $(1 - f_\alpha^{234}) \times (1 - f_\alpha^{230})$); and $^{10}$Be exposure ages. Symbol colours represent the third parameter in each plot. Sorting observed in the symbol colours helps to explain the noise associated with the other two parameters. In particular, the scatter in the decrease of the activity ratios over time can be largely explained by the variability in the recoil loss factor (top left plot). The plots also let us assume that the variability in the weathering coefficients could be low, otherwise a residual dispersion would be observed. In all cases, error bars are smaller than the size of the dots.





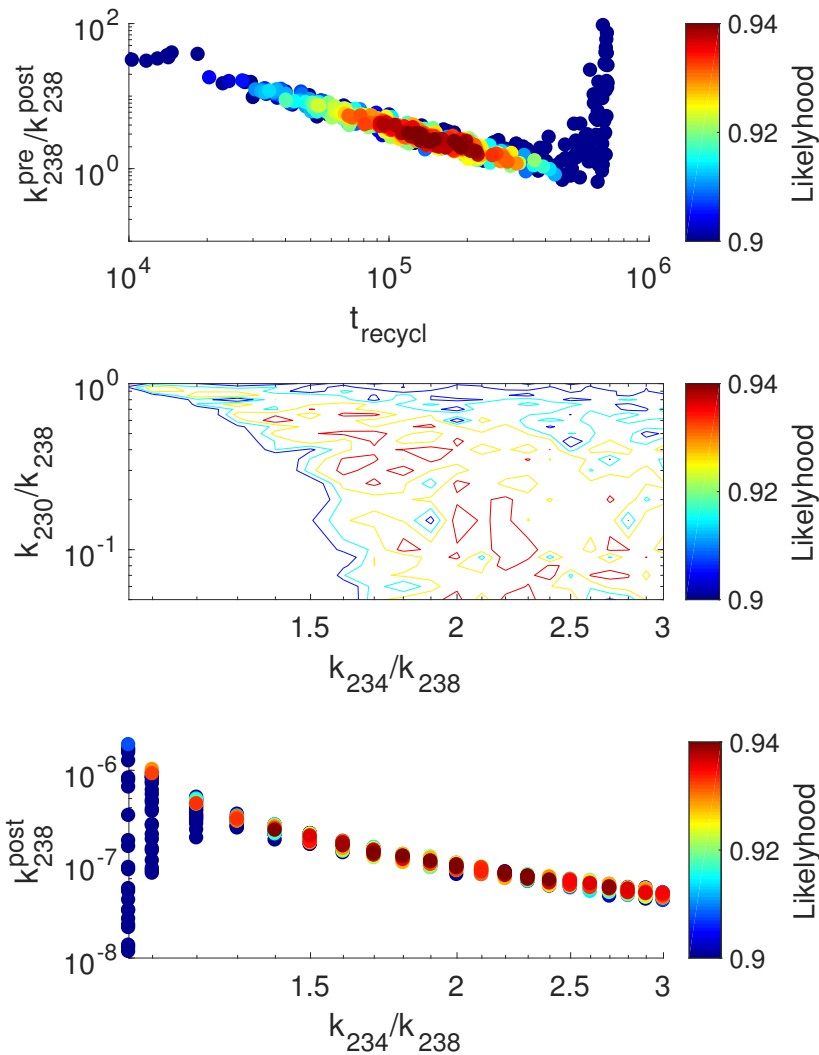

**Figure 6.** Results of Monte Carlo simulations for the U-Th data. The same Monte Carlo simulations were conducted for different fixed values of $k_{234}/k_{238}$ and $k_{230}/k_{238}$. For each simulation we estimated the optimal solution for $k_{238}^{pre}$, $k_{238}^{post}$ and recycling time (pre-deposition duration, after comminution), i.e. the solution with the lowest misfit between measured and calculated activity ratio data. We used this misfit to calculate a likelihood between 0 and 1, shown by the colourmap (red is lower misfit, blue is higher misfit, stretched to fit our results). The data suggest that silts experienced stronger weathering before deposition. (Upper panel) The stronger the weathering, the lower the pre-depositional duration needs to be to fit the data. The optimal solution is for three times stronger weathering before deposition and a recycling time of 100 ka. (Middle panel) We poorly constrain the relative weathering intensities of $^{234}$U, $^{238}$U and $^{230}$Th, but a relationship can be extracted between their ratios. (Lower panel) Another challenge in constraining the pre-depositional duration is the dependence between weathering intensity and relative weathering intensity of $^{234}$U compared to $^{238}$U. The estimation of sediment transfer times using U-Th disequilibria series could be improved with better constraints on the relative mobility of these two isotopes.

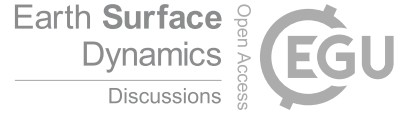

**Table 1.** Cosmogenic [10]Be nuclide data for outwash surface cobbles and depth profile samples. Moreno 1 and 3 data are already published in Hein et al. (2017).

| Sample ID | Latitude (DD) | Longitude (DD) | Elev. (m asl) | Depth (cm) | Shielding correction | Thickness (cm) | [10]Be ± 1$\sigma$ ($10^5$ at.g$^{-1}$) |
|---|---|---|---|---|---|---|---|
| *Moreno 1 outwash surface cobbles* | | | | | | | |
| M1-T5 | -46.5544 | -70.8758 | 496 | - | 1 | 7 | 10.75 ± 0.21 |
| M1-T9 | -46.5544 | -70.8758 | 496 | - | 1 | 6.5 | 12.64 ± 0.23 |
| M1-T12 | -46.5544 | -70.8758 | 496 | - | 1 | 7 | 15.59 ± 0.28 |
| M1-T13 | -46.5544 | -70.8758 | 496 | - | 1 | 6 | 15.09 ± 0.27 |
| *Moreno 3 outwash surface cobbles* | | | | | | | |
| M3-T6 | -46.547 | -70.8268 | 482 | - | 1 | 7 | 10.14 ± 0.31 |
| M3-T11 | -46.547 | -70.8268 | 482 | - | 1 | 7 | 14.53 ± 0.26 |
| M3-T14 | -46.547 | -70.8268 | 482 | - | 1 | 4 | 13.36 ± 0.36 |
| M3-T16 | -46.547 | -70.8268 | 482 | - | 1 | 5 | 16.07 ± 0.29 |
| *Deseado 1 outwash surface cobbles* | | | | | | | |
| D1-T1 | -46.5219 | -70.7732 | 522 | - | 1 | 4 | 25.75 ± 0.54 |
| D1-T3 | -46.5219 | -70.7732 | 522 | - | 1 | 7 | 27.10 ± 0.53 |
| D1-T12 | -46.5219 | -70.7732 | 522 | - | 1 | 6.5 | 26.11 ± 0.47 |
| *Deseado 2 outwash surface cobbles* | | | | | | | |
| D2-T5 | -46.5140 | -70.7364 | 535 | - | 1 | 3 | 31.09 ± 0.57 |
| D2-T6 | -46.5140 | -70.7364 | 535 | - | 1 | 3 | 36.16 ± 0.65 |
| D2-T12 | -46.5140 | -70.7364 | 535 | - | 1 | 3.5 | 18.50 ± 0.34 |
| *Deseado 2 outwash depth profile* | | | | | | | |
| D2-50 | -46.5015 | -70.7368 | 544 | 50 | 1 | 4 | 15.23 ± 0.28 |
| D2-60 | -46.5015 | -70.7368 | 544 | 60 | 1 | 4 | 11.97 ± 0.25 |
| D2-70 | -46.5015 | -70.7368 | 544 | 70 | 1 | 4 | 10.36 ± 0.21 |
| D2-130 | -46.5015 | -70.7368 | 544 | 130 | 1 | 4 | 4.65 ± 0.18 |
| D2-230 | -46.5015 | -70.7368 | 544 | 230 | 1 | 4 | 1.40 ± 0.07 |
| *Telken 5 outwash depth profile* | | | | | | | |
| TK-50 | -46.4439 | -70.5069 | 552 | 50 | 1 | 4 | 16.65 ± 0.37 |
| TK-60 | -46.4439 | -70.5069 | 552 | 60 | 1 | 4 | 15.92 ± 0.29 |
| TK-70 | -46.4439 | -70.5069 | 552 | 70 | 1 | 4 | 14.20 ± 0.28 |
| TK-130 | -46.4439 | -70.5069 | 552 | 130 | 1 | 4 | 5.13 ± 0.30 |
| TK-240 | -46.4439 | -70.5069 | 552 | 240 | 1 | 4 | 1.94 ± 0.12 |





**Table 2.** Modeled exposure ages for outwash surface cobbles and depth profile samples (see main text and Supplementary Information for details of profile modeling). Moreno 1 and 3 ages have been already publised in Hein et al. (2017).

| Sample ID | Age (ka) | External error (ka) | Profile age (ka) | Upper error (ka) | Lower error (ka) |
|---|---|---|---|---|---|
| *Moreno 1 outwash surface cobbles* | | | | | |
| M1-T5 | 177 | 17 | | | |
| M1-T9 | 208 | 21 | | | |
| **M1-T12** | **261** | **26** | | | |
| M1-T13 | 250 | 25 | | | |
| *Moreno 3 outwash surface cobbles* | | | | | |
| M3-T6 | 168 | 17 | | | |
| M3-T11 | 246 | 25 | | | |
| M3-T14 | 219 | 22 | | | |
| **M3-T16** | **269** | **27** | | | |
| *Deseado 1 outwash surface cobbles* | | | | | |
| D1-T1 | 430 | 46 | | | |
| **D1-T3** | **468** | **50** | | | |
| D1-T12 | 446 | 47 | | | |
| *Deseado 2 outwash surface cobbles* | | | | | |
| D2-T5 | 520 | 56 | | | |
| **D2-T6** | **618** | **68** | | | |
| D2-T12 | *293* | 30 | | | |
| *Deseado 2 outwash depth profile* | | | | | |
| | | | 600 | 70 | 35 |
| *Telken 5 outwash depth profile* | | | | | |
| | | | 780 | 170 | 80 |





**Table 3.** U-Th concentrations, activity ratios and BET data (specific surface areas and fractal dimensions, and calculated recoil loss factors). Duplicates are also shown, two for the whole process (including sieving, leachings, HF digestion, chromatography), and one for HF digestion and chromatography.

| sample | moraine | Latitude | Longitude | [U] (ppm) | 2SE | [Th] (ppm) | 2SE | Specific surface area (m2.g-1) | Fractal dimension | $f_\alpha^{234}$ |
|---|---|---|---|---|---|---|---|---|---|---|
| EX-1 | glacial flour | -46.5026 | -73.1642 | 4.186 | 0.010 | 22.70 | 0.06 | 1.051 | 2.49 | 0.0047 |
| F1-U-M2 | Fenix 1 | -46.5974 | -71.0359 | 1.680 | 0.002 | 6.36 | 0.01 | 1.831 | 2.52 | 0.0075 |
| M1-U-M | Moreno 1 | -46.5587 | -70.8883 | 1.855 | 0.003 | 7.63 | 0.01 | 1.392 | 2.46 | 0.0069 |
| M3-U-M | Moreno 3 | -46.5574 | -70.8494 | 2.148 | 0.003 | 8.67 | 0.02 | 4.719 | 2.56 | 0.0170 |
| D1-U-M | Deseado 1 | -46.5255 | -70.7837 | 1.928 | 0.004 | 8.00 | 0.02 | 2.238 | 2.53 | 0.0089 |
| D2-U-M | Deseado 2 | -46.5170 | -70.7518 | 1.503 | 0.002 | 5.94 | 0.01 | 3.535 | 2.53 | 0.0140 |
| TI-U-M | Telken 1 | -46.4904 | -70.7240 | 1.633 | 0.003 | 6.56 | 0.01 | 2.081 | 2.45 | 0.0107 |
| TK-U-M | Telken 5 | -46.4552 | -70.5292 | 1.593 | 0.003 | 6.96 | 0.02 | 5.435 | 2.57 | 0.0190 |
| *Duplicates* | | | *Observation* | | | | | | | |
| M3-U-M 2 | duplicate of the whole process | | | 2.067 | 0.004 | 9.38 | 0.02 | 7.034 | 2.59 | 0.022 |
| M3-U-M 2 bis | duplicate from the HF digestion step | | | 1.990 | 0.003 | 8.52 | 0.02 | | | |
| TI-U-M 2 | duplicate of the whole process | | | 1.776 | 0.003 | 6.92 | 0.01 | 1.818 | 2.54 | 0.0070 |

| sample | moraine | $^{238}U/^{232}Th$ | 2SE | $^{234}U/^{238}U$ | 2SE | $^{230}Th/^{238}U$ | 2SE | $^{230}Th/^{234}U$ | 2SE |
|---|---|---|---|---|---|---|---|---|---|
| EX-1 | glacial flour | 0.561 | 0.002 | 1.007 | 0.001 | 1.022 | 0.005 | 1.015 | 0.005 |
| F1-U-M2 | Fenix 1 | 0.803 | 0.002 | 0.968 | 0.001 | 0.998 | 0.005 | 1.031 | 0.006 |
| M1-U-M | Moreno 1 | 0.739 | 0.001 | 0.971 | 0.001 | 0.970 | 0.004 | 0.999 | 0.004 |
| M3-U-M | Moreno 3 | 0.753 | 0.002 | 0.959 | 0.001 | 0.962 | 0.004 | 1.003 | 0.004 |
| D1-U-M | Deseado 1 | 0.733 | 0.002 | 0.968 | 0.001 | 0.965 | 0.004 | 0.998 | 0.005 |
| D2-U-M | Deseado 2 | 0.769 | 0.001 | 0.956 | 0.001 | 0.943 | 0.004 | 0.987 | 0.004 |
| TI-U-M | Telken 1 | 0.757 | 0.002 | 0.958 | 0.001 | 0.960 | 0.005 | 1.002 | 0.005 |
| TK-U-M | Telken 5 | 0.696 | 0.002 | 0.947 | 0.001 | 0.937 | 0.004 | 0.989 | 0.005 |
| *Duplicates* | | | | | | | | | |
| M3-U-M 2 | | 0.671 | 0.002 | 0.955 | 0.001 | 0.964 | 0.006 | 1.009 | 0.006 |
| M3-U-M 2 bis | | 0.711 | 0.002 | 0.955 | 0.001 | 0.962 | 0.004 | 1.006 | 0.005 |
| TI-U-M 2 | | 0.780 | 0.001 | 0.963 | 0.001 | 0.964 | 0.005 | 1.001 | 0.006 |