# Peer review of "U-Th and 10Be constraints on sediment recycling in proglacial settings, Lago Buenos Aires, Patagonia"

_Earth Surface Dynamics, 2017_

## Referee Comment (RC1) · Anonymous Referee #1 · 1 Oct 2017

This is a very interesting and appropriate paper for ESD. There are enough new 10Be data and as far as I know (and the authors know) it is the 1st U-Th data and application in southern South American on sediment movement from source to sink, as they say. Hence, this is a novel contribution. I think my comments are moderate to major and not difficult to do.

Some major (or moderate?) comments.

1. One of my most significant comments is that one really needs to be an expert in U-Th series measurements and approach to really follow details of the Methods and Results (not so much Discussion). So much of it may remain largely inaccessible to

the non-expert, which is a shame, given all their hard-work!!

I am actually not sure how much they can do about this, as the paper is not an appropriate place for a 101 lesson on the U-Th comminution and sediment approaches. On the other hand, I think there are few minor things they can do which I suggest or highlight below (e.g., see figure captions) which will at least help the reader a bit not familiar with these methods. These mainly focus around i) explain figure captions a bit better and ii) perhaos more info in supplement information section (a user friendly flow chart or Table of terms, as suggested below?) iii) some more labels on the figures (see below) and iv) a few places in text can add a few more words (see below)

I should also highlight to the Editor that I am not an expert on the U-Th series and application. I cannot really evaluate this in detail.

2. Some more info is needed on setting and context of samples. i) Specifically some photos of the samples so reader can see the context. This includes photos of the cosmo samples. If possible, they need to provide a photo of the outwash with the moraine behind it (showing how they link); if they do not have such a photo, maybe the closest thing they have. At least one or two photos of the silts they sampled for comminution – are these from a pit? A road cut? No detailed geologic context is provided to the reader to evaluate their results. ii) I think also a more zoomed in version of the map over the sampling area. It is hard to appreciate from their short discussion. See comments below also for Figure 1. iii) Can they provide a strat section, even schematic or simplified?

Minor - also, on map – show Deseado boulder location, since mentioned in the text.

3) Also, I recall there were two telken boulders in Kaplan et al. which were used to derive maximum erosion rates – they are very maximum erosion ages, and very minimum exposure dates, but the authors may want to mention (?) these data because they support i) need to date outwash on these old deposits as they do (and Hein et al do) and ii) the erosion rate they derive below shows these Telken bouder measurements were

indeed maximum erosion rates as Kaplan et al. 2005 stated. More on this below.

4. Apparently, Douglass had measured Deseado boulders in his PhD research that are unpublished, except for abstracts. One abstract that is particularly relevant that authors may want to know about, is a GSA abstract with recalculated erosion rates; these are close to what they derive on the outwash I recall. The abstract is cited in Hein et al 2017. They may want to refer to this, as it is another measurement that supports their finding for very low erosion rates in the area, much lower than the 'max rates' presented in Kaplan et al. 2005. The reference (can be looked up) is GSA Northeastern Section - 42nd Annual Meeting (12-14 March 2007) Paper No. 24-2: CONSTRAIN-ING BOULDER EROSION RATES AND AGES OF MID-PLEISTOCENE MORAINES, LAGO BUENOS AIRES, ARGENTINA (again see Hein paper for references).

5. Can they say if there are any changes in the U-series data over the time of the moraines and sediments listed in Table 3? They sampled a nice profile or transect from old to young moraines, spanning almost 1 million years; this is unique aspect of their study (has this been done even anywhere?). They do not really discuss this at all (unless I missed it). Is there a lot more information here they can highlight or speculate about? For example, are the U-Th results different for moraines from stage 8 and 2, compared with their data on the older telken and deseado sediments Or vice versa? A point is that I am wondering if they can elucidate changes as Southern Andes are eroding down (they discuss Kaplan et al. 2009 mechanism, for example, in this context).E.g., glacial erosive products produced and recycled more and more. . .?

Other, detailed comments corresponding to pages (and providing more info on above comments):

page (p.) 2 line 1 – what do they mean by 'remobilization of moraines' - this sounds catastrophic ! Please rephrase to be more precise in the wording. Do they mean remobilization of fine sediments in matrix? Material they sampled/measured? Moraine matrix?

p. 3. For the part of the study dealing with sediment measurement and findings, it may be of interest to mention that ice sheet glaciations started around 7 to 5 Ma (Mercer and Sutter (1982 i recall)) with tills and supposedly moraines and ice sheet glacial erosion, for several million years, even before the Telken stuff deposited (are there implications for history of fines, glacial rock flour etc?)

line 23 "finally we are able.....sediment history is likely..." I would clarify what kind of sediment – fine sediment? For example, some would include boulders in this context, as the statement is written. I think they mean the type of materials they are working on – if not, their data only apply to the type of sediment sizes they are working on (e.g., rock flour).

p. 4. There is another moraine, the Menucos moraine (Singer et al., 2004; Douglass et al. 2006). Kaplan et al. (2011) recalculated these ages.; they are albeit buried in the last table of the 2011 supplement. The key point is the recalculated ages are around 17 ish (18 to 17 ka), I recall, ignoring obvious outliers.

Line 18 to 20. A clear example in which some photos (2 for example ) would of immense help for evaluating what they sampled. A pit? A road cut? Can they provide a strat section, even schematic or simplified?

p.5 see comment about photos

p.6 lines 14+ 1) say what the 'global rate' of Borchers et al rate is at SLHL. 2) how does global rate of Borchers et al compares with the local rate in Kaplan et al. 2011 and whether the difference in rates will mean for their older results. For younger features, should not matter too much, but once you get to Deseado and Telken age, even small differences between Borchers et al and Kaplan et al. rates could be significant given non-linear age equation.

I bring this up – in part – because I am actually wondering if they will get slightly better agreement with Ar/Ar and/or marine oxygen isotope comparisons; I cannot recall exact

**ESurfD**
**s, but I recall the Borchers et al. rate is higher than that in Kaplan et al. which is for Patagonia. It is easy for the authors to estimate the difference – on the online cronus calculator Balco provides 'alternative versions for 2.2. Although still v.2.2 it is good enough to see the effect of using a slightly lower PR (Kaplan et al. 2011), if correct.**

p. 7 to 11 – see comment above about U-Th analyses.

p. 11 section 2.3.4. if they can give one sentence on why this matters, to the non-expert, that would be helpful.

p. 13. As mentioned above, it is relevant to mention the Douglass et al study, albeit just in GSA abstract form, which has low erosion rates on par with their values. It is important for them to point out that this is much lower (and more realistic) than that boulder erosion rate presented in Kaplan et al 2005, which presented maximum rates. Albeit Kaplan et al. 2005 and douglass et al were for moraine boulders (may be slightly different than their data given possibly more exhumation on the moraines).

p. 14 line 8 and 9 (last sentence before new section). Regarding the sensitivity tests, I suggest they may want to mention that the global rate in Borchers et al vs local rate in Kaplan et al., although very small, does not make a difference given a non-linear age equation. If the "global" rate in Borchers et al. is slightly higher (I recall), which is what is listed on the Cronus 2.3 site. The importance is that with a higher rate they get slightly lower ages, especially for Deseado and Telken time.

p. 15. Line 13 Section 3.3 title For non-experts, what do they mean by ': a 3-dimensional problem"

This is another example how a little bit more info may make these aspects of the paper a little more accessible to the general ESD audience.

Line 16. What do they mean by youngest moraine sample – which sample (refer to which sample in table)?

p. 16. glacial periods are much dustier – with likely much of the dust coming from

Patagonia (lots of literature on this). Perhaps 1 or 2 sentences on what such dust inputs may be bringing in, in the context of their results. Is this another implication they can highlight of their U-Th results and findings?

p. 17. Line 18. Add something along the lines of "slight changes in scaling factors (see methods) do not affect this finding."

Line 18 – the 476 ka – is this recalculated? I assume it is from the 2005 paper, but please state so.

Line 24 before "Smedley" a word or two missing or open parentheses.

p. 18. Top line 1 – spelling after the word Telken Line 27 grammar problem

p. 19 Line 14-15. First two sentences - awkward writing.

p. 20. per comment above, sorry, perhaps my non-expertise, but, can the authors say if there is a change in the U-Th data between Telken and Fenix moraines. A change in time of weathering rates? The relevance of the issue is testing the idea they discuss, in Kaplan et al. 2009. Is there a change in results and thus sediment transfer times/comminution age over the 1 Ma as valleys are excavated and Andes erode down?

Lines 17-19. Starting with "this is exacerbated. . ." I do not understand this sentence – it is not clear. The authors need to clarify.

Line 23. What do they mean by 'moraine sediment.' That is very broad. Does this also apply to boulders? Or just find matrix in moraines? Please clarify (a similar question was raised earlier about the whole moraine being remobilized).

p. 21. Line 24. I think I know what the authors mean, but as written, not recorded or not preserved is same thing. Rephrase to, not recorded because not preserved, or do they mean not preserved or ice did not reach this far? Something similar?

Table 1. It is becoming customary to also provide AMS ratios and carrier added (9Be).

Given that (I think) it is standard enough to do this now in cosmo literature, that I have to insist these two columns be added to the table.

Also, in caption, please remind reader of AMS standard used, given it is not common (except at ETH).

FIGURES Figure 1. It is difficult to evaluate fully the morphostratigraphic relations that are essential for interpretation of data from this figure: 1) need some photos associated with these maps – see comments above. This is especially so for the outwash linked to the moraine – cannot appreciate this from the map. Otherwise need to say perhaps not directly related to the specific moraine – does it actually really matter for their main findings if they establish this? that is, deseado outwash is deseado outwash whether specifically linked to a specific crest in Singer et al, 2004. 2) I suggest one more panel really zooming in on the key area where the crests, outwash, and samples are. A panel D. This could be in the SOM if there is not enough room in the paper. The photos could also be in the SOM – just add one figure of several photos.. Any strat sections, with where the samples are from, that they can provide in SOM? I think it is always best to have the photos in the main text if there is space. However, if there is not enough space, at least to have them in the supplement info as a figure. 3) explain CDI and SPI in caption. SPI is obvious, but not CDI ? there are lots of glaciers in C. Darwin– what is specifically being highlighted and why for CD? the biggest ice mass in CD? 4) there is an error in the caption – the transect is in panel B (it says panel C, but this is in B)?. 5) after "[c]": add "[c] Inset shows a west-east transect….." 6) For cosmo ages, it would better to make the symbols the same colors as those on figure 2. Easier for the reader not familiar with the area. I would also make the 2 outwash symbols as squares on Figure 1, just like on figure 2. 7) plot Deseado boulder that you refer to, from the literature (is from Kaplan et al. 2005 ?)

Figure 2 1) maybe also plot here the Deseado boulder age (without erosion) that refer to? If do plot it, say it is a minimum age. Up to authors though whether they think it is best to plot the boulder cosmo age on figure 2, or it would be too messy. It would

highlight again the moraine oulder ages too young. 2) Could all Deseado 2 ages be minima? Not just the 293? Deseado 1 has such a nice tight cluster, hard to envision being too minimum, but Deseado 2 is more scattered and only 2 ages. 3) I suggest showing the full LR 2005 curve. It is relatively easy to do, it does not take up much room and would be more informative than showing positive Stages. It would convey more info such as stage 8 not being a big global glacial maxima (cf., Hein et al.) 4) in caption, say what the effect is of using local PR versus global PR of Borchers et al, if significant ? see above comment?. Although not much for younger moraines, given the age equation is non-linear, it can add up for older Deseado moraines?

Figure 3. 1) This plot is difficult to digest for the broader readership of ESD. Perhaps add some labels. For example, in the bottom 3 quadrants, can they add labels to all or some of the quadrants to summarize what we are looking at? In the top panels maybe also some more labels would help – what is the blue line? What are the red lines? It is explained in the caption, but, per my comment at top of review I am just trying to think of ways to make this figure more accessible to all readers.

2) Also, it is not clear – are these all hypothetical – or based on the monte carlo simulations – it is implied so ('results' mentioned in caption), but it is not specifically stated, and obvious to non-expert. 3) 4th line from bottom – 10Be ages – please explain which 10Be ages – reader cannot tell simply from the figure. 4) third line from bottom – "Weathering intensities are lower after moraine deposition" – how do you know this? That is, please explain from the plot how this is known. Or show visually with suggestion above to add labels.

Figure 6. Needed to explain a little better for non-experts – see comments at start of review. For non-experts, what is likelihood signify? Instead of Trecycle – spell out? Maybe a table in the SOM of all of these terms with a brief explanation, for non-experts? In panel A for example, they may add some info to the plot itself – some labels of what we are looking at. For example, what does the bend and dark vertical-aligned purple dots mean? Maybe a few labels on this plot and the bottom plot would make this more

accessible to nonexperts in U-Th approach.

Supplement information.

Section 2. Given the prominent role the profiles and monte carlo simulation play in the paper - this section needs to be expanded to explain: 1) what the monte carlo optimization is and how it is done? Some of this is in the main text, but more details in the context of the Supplement are appropriate here in my opinion. 2) a paragraph (at least) explaining what we are looking at on Figures 1 and 2. What are these panels? The caption should refer to each panel. Or the text. Please explain each panel in more depth and/or in caption. The right panel – this is a key figure. Explain more – what is reader looking at? Not much is needed, perhaps just a longer caption, but more is warranted in explaining Figures 1 and 2 in the context of supplementary information and simply to describe what the reader is looking at, which are some of the key results discussed in the main text.

Figure 3 – in caption, for non-expert, can they add one more sentence saying what f230 = 0 means? The authors also may want to add some labels along the lines of the comment above for main text figures.

Figure 5. Explain specifically where time comes from (e.g., U-Th dates associated with moraine/outwash dating....and so on, see table 3)

Per comments above, add photos and any strat sections. I would put the photos in the main text if there is room.

---

## Referee Comment (RC2) · Anonymous Referee #2 · 5 Oct 2017

This manuscript present some interesting cosmogenic and U-series isotope data. While the combination is interesting, it seems that isotopic systems are used independently with little overlap. Cosmogenic isotopes were used to date the moraines on the one hand, and U-series to infer information about sediment transport on the other hand. It almost reads like 2 independent manuscripts. While I command the authors' efforts, and this work should of course eventually be published, there are many aspects that the authors may want to consider, and which significantly affect how the U-series data are interpreted. I provide detailed comments in the PDF attached but the main points are that: - the fractal correction is possibly not needed and result in an underestimation of recoil fractions. As a result, the authors explore a more complex

(but less constrained) model (previously proposed by Scott et al. in the early 1990's, but also more recently re-introduced). A simple way to test this is to look at the type of isotherms obtained during BET analysis. Only a mesoporous would require the fractal correction to be invoked. I think Lee investigated this but as far as I know this is unpublished, but the authors may want to consider this point. - while the weathering model is presented, there is a lack of details on how it is solved, and what justifies the assumption made in the model (e.g. same 'recycling time' and weathering intensity for all samples? why? this record cover a very variable period of Earth's history). I only recommend rejection because I believe the extent of revisions to be undertaken is large, but a new manuscript should be submitted, taking into consideration the comments above and in the document attached.

Please also note the supplement to this comment:
https://www.earth-surf-dynam-discuss.net/esurf-2017-45/esurf-2017-45-RC2-supplement.pdf

―――――――――――――――――――――

**Supplement:**

[revised manuscript text omitted]

Earth **Surface**
**Dynamics**
Discussions

---

## Author Comment (AC1) · 22 Nov 2017

Dear editor, dear reviewers,

We have now completed a revision of our manuscript "U-Th and 10Be constraints on sediment recycling in proglacial settings, Lago Buenos Aires, Patagonia". We would like to acknowledge the interesting and exhaustive reviews by the two reviewers, that helped us improve the manuscript. We did our best to address all comments and suggestions.

We address our answer to more specific comments in the attached file. The manuscript

has been improved following these remarks. We hope that both will help clarifying various addressed points.

Best regards Antoine Cogez

Please also note the supplement to this comment:
https://www.earth-surf-dynam-discuss.net/esurf-2017-45/esurf-2017-45-AC1-supplement.pdf

---

## Author Comment (AC2) · 22 Nov 2017

Dear editor, dear reviewers,

We have now completed a revision of our manuscript "U-Th and 10Be constraints on sediment recycling in proglacial settings, Lago Buenos Aires, Patagonia". We would like to acknowledge the interesting and exhaustive reviews by the two reviewers, that helped us improve the manuscript. We did our best to address all comments and suggestions.

Below we address our answer to more specific comments. The manuscript has been improved following these remarks. We hope that both will help clarifying various addressed points.

Best regards

Antoine Cogez

**Anonymous Referee #1**

This is a very interesting and appropriate paper for ESD. There are enough new 10Be data and as far as I know (and the authors know) it is the 1st U-Th data and application in southern South American on sediment movement from source to sink, as they say. Hence, this is a novel contribution. I think my comments are moderate to major and not difficult to do.

Some major (or moderate?) comments.

1. One of my most significant comments is that one really needs to be an expert in U-Th series measurements and approach to really follow details of the Methods and Results (not so much Discussion). So much of it may remain largely inaccessible to the non-expert, which is a shame, given all their hard-work!!

I am actually not sure how much they can do about this, as the paper is not an ap- propriate place for a 101 lesson on the U-Th comminution and sediment approaches. On the other hand, I think there are few minor things they can do which I suggest or highlight below (e.g., see figure captions) which will at least help the reader a bit not familiar with these methods. These mainly focus around i) explain figure captions a bit better and ii) perhaos more info in supplement information section (a user friendly flow chart or Table of terms, as suggested below?) iii) some more labels on the figures (see below) and iv) a few places in text can add a few more words (see below)

I should also highlight to the Editor that I am not an expert on the U-Th series and application. I cannot really evaluate this in detail.

We recognize that U-Th series disequilibrium is conceptually difficult to grasp and it is difficult making it clear in a new paper about the topic. We added a section in suplementary materials in order to make it clearer. We hope the basic understanding can be achieved in the paper, and that this added section helps to orient the reader to find more detailed informations about U series. Moreover we address your different detailed points below in the list of precise comments you

addressed, which helped improving the clarity of those points.

2. Some more info is needed on setting and context of samples. i) Specifically some photos of the samples so reader can see the context. This includes photos of the cosmo samples. If possible, they need to provide a photo of the outwash with the moraine behind it (showing how they link); if they do not have such a photo, maybe the closest thing they have. At least one or two photos of the silts they sampled for comminution – are these from a pit? A road cut? No detailed geologic context is provided to the reader to evaluate their results. ii) I think also a more zoomed in version of the map over the sampling area. It is hard to appreciate from their short discussion. See comments below also for Figure 1. iii) Can they provide a strat section, even schematic or simplified? Minor - also, on map – show Deseado boulder location, since mentioned in the text.

Some pictures were added in the suplementary materials.

3) Also, I recall there were two telken boulders in Kaplan et al. which were used to de- rive maximum erosion rates – they are very maximum erosion ages, and very minimum exposure dates, but the authors may want to mention (?) these data because they sup- port i) need to date outwash on these old deposits as they do (and Hein et al do) and ii) the erosion rate they derive below shows these Telken bouder measurements were indeed maximum erosion rates as Kaplan et al. 2005 stated. More on this below.

It is indeed interesting that both our estimated deflation rate and the boulder erosion rates from Kaplan (2005) are comparable (maximum values at least). However can we compare boulder erosion rates and outwash surface deflation ? We prefer not mentioning it since we remain unsure both can actlly be compared. It could blur the understanding about the interest of taking outwash cobbles instead of moraines boulders.

4. Apparently, Douglass had measured Deseado boulders in his PhD research that are unpublished, except for abstracts. One abstract that is particularly relevant that au- thors may want to know about, is a GSA abstract with recalculated erosion rates; these are close to what they derive on the outwash I recall. The abstract is cited in Hein et al 2017. They may want to refer to this, as it is another measurement that supports their finding for very low erosion rates in the area, much lower than the 'max rates' pre- sented in Kaplan et al. 2005. The reference (can be looked up) is GSA Northeastern Section - 42nd Annual Meeting (12-14 March 2007) Paper No. 24-2: CONSTRAIN- ING BOULDER EROSION RATES AND AGES OF MID-PLEISTOCENE MORAINES, LAGO BUENOS AIRES, ARGENTINA (again see Hein paper for references).

Same comment than above, we might be wrong but we prefer not mentioning it since we are not sure the comparison is appropriate.

5. Can they say if there are any changes in the U-series data over the time of the moraines and sediments listed in Table 3? They sampled a nice profile or transect from old to young moraines, spanning almost 1 million years; this is unique aspect of their study (has this been done even anywhere?). They do not really discuss this at all (unless I missed it). Is there a lot more information here they can highlight or speculate about? For example, are the U-Th results different for moraines from stage 8 and 2, compared with their data on the older telken and

deseado sediments Or vice versa? A point is that I am wondering if they can elucidate changes as Southern Andes are eroding down (they discuss Kaplan et al. 2009 mechanism, for example, in this context).E.g., glacial erosive products produced and recycled more and more. . .?

One assumption of the model we use is that the recycling time is the same for all samples, so we cannot discuss a trend in the evolution of the recycling time over the last million year. First, given the number of uncertainties in the problem, we need to solve a single recycling time. Second we can argue that this hypothesis of a constant recycling time over the last million year remains reasonable : see our response to the third main point of reviewer 2 for a more detailed discussion on these explanations. The main point is that the glacial valleys and the overdeepening (i.e. the main sediment sinks, responsible for increasing the sediment transport time) were most probably already carved by 1 Ma. However it appears that this was unclear, so we tried to make clearer in the revised version of the manuscript by adding a paragraph in section 2.3.5.

Other, detailed comments corresponding to pages (and providing more info on above comments):

page (p.) 2 line 1 – what do they mean by 'remobilization of moraines' - this sounds catastrophic ! Please rephrase to be more precise in the wording. Do they mean remobilization of fine sediments in matrix? Material they sampled/measured? Moraine matrix?

We changed to 'remobilization of sediments from moraines…'

p. 3. For the part of the study dealing with sediment measurement and findings, it may be of interest to mention that ice sheet glaciations started around 7 to 5 Ma (Mercer and Sutter (1982 i recall)) with tills and supposedly moraines and ice sheet glacial erosion, for several million years, even before the Telken stuff deposited (are there implications for history of fines, glacial rock flour etc?)

We added « Mercer and Sutter (1982) showed that the oldest glaciogenic sediments we find in the area are 6 Ma, and consist of tills interbedded with lava flows that preserved them from erosion », in the section settings

line 23 "finally we are able. . ...sediment history is likely. . ." I would clarify what kind of sediment – fine sediment? For example, some would include boulders in this context, as the statement is written. I think they mean the type of materials they are working on – if not, their data only apply to the type of sediment sizes they are working on (e.g., rock flour).

Ok, added « fine » to make it clearer

p. 4. There is another moraine, the Menucos moraine (Singer et al., 2004; Douglass et al. 2006). Kaplan et al. (2011) recalculated these ages.; they are albeit buried in the last table of the 2011 supplement. The key point is the recalculated ages are around 17 ish (18 to 17 ka), I recall, ignoring obvious outliers.

These moraines have their importance in the area, however our objective is to constrain the U-series measurements with the 10Be data, so we feel it was outside the scope of the study.

Line 18 to 20. A clear example in which some photos (2 for example ) would of immense help for evaluating what they sampled. A pit? A road cut? Can they provide a strat section, even schematic or simplified?

Some pictures were added in the supplementary materials.

p.5 see comment about photos

Changed

p.6 lines 14+ 1) say what the 'global rate' of Borchers et al rate is at SLHL. 2) how does global rate of Borchers et al compares with the local rate in Kaplan et al. 2011 and whether the difference in rates will mean for their older results. For younger features, should not matter too much, but once you get to Deseado and Telken age, even small differences between Borchers et al and Kaplan et al. rates could be significant given non-linear age equation.

I bring this up – in part – because I am actually wondering if they will get slightly better agreement with Ar/Ar and/or marine oxygen isotope comparisons; I cannot recall exact #s, but I recall the Borchers et al. rate is higher than that in Kaplan et al. which is for Patagonia. It is easy for the authors to estimate the difference – on the online cronus calculator Balco provides 'alternative versions for 2.2. Although still v.2.2 it is good enough to see the effect of using a slightly lower PR (Kaplan et al. 2011), if correct.

Thanks for making this point. We mention l. 21 p. 6 that changing the site production rate of +/- 0.3 at/g/yr does not significatively change the results. This range of site productions rates corresponds to changing the SLHL production rate from 3.6 to 4 at/g/yr. 3.6 is the minimum production rate at SLHL in Kaplan et al. (2011) and 4 is the production rate at SLHL in Borchers et al (2016), both using the Lm scaling scheme. We already tested what is mentioned by the reviewer. However, it was not clear in the paper. So we added two sentences in paragraph 2.2.3 to precise this point.

p. 7 to 11 – see comment above about U-Th analyses.

p. 11 section 2.3.4. if they can give one sentence on why this matters, to the non- expert, that would be helpful.

The recoil loss factor is a key parameter for the determination of the comminution age. This parameter and the equations are presented in the section 2.3.1. In section 2.3.4, we present the method we used to estimate it. We added a few sentence in section 2.3.1.

p. 13. As mentioned above, it is relevant to mention the Douglass et al study, albeit just in GSA abstract form, which has low erosion rates on par with their values. It is important for them to point out that this is much lower (and more realistic) than that boulder erosion rate presented in Kaplan et al 2005, which presented maximum rates. Albeit Kaplan et al. 2005 and douglass et al were for moraine boulders (may be slightly different than their data given possibly more exhumation on the moraines).

See our response above

p. 14 line 8 and 9 (last sentence before new section). Regarding the sensitivity tests, I suggest they may want to mention that the global rate in Borchers et al vs local rate in Kaplan et al., although very small, does not make a difference given a non-linear age equation. If the "global" rate in Borchers et al. is slightly higher (I recall), which is what is listed on the Cronus 2.3 site. The importance is that with a higher rate they get slightly lower ages, especially for Deseado and Telken time.

Given your previous comment, this point is addressed in section 2.2.3.

p. 15. Line 13 Section 3.3 title For non-experts, what do they mean by ': a 3- dimensional problem"

This is another example how a little bit more info may make these aspects of the paper a little more accessible to the general ESD audience.

This expression '3 dimensional problem' is misleading, we removed it from the title of the section.

Line 16. What do they mean by youngest moraine sample – which sample (refer to which sample in table)?

The youngest moraine is the sample F1. We modified the text accordingly.

p. 16. glacial periods are much dustier – with likely much of the dust coming from Patagonia (lots of literature on this). Perhaps 1 or 2 sentences on what such dust inputs may be bringing in, in the context of their results. Is this another implication they can highlight of their U-Th results and findings?

Your point is indeed very interesting. We discuss it at the beginning of section 4.2 (first paragraph).

p. 17. Line 18. Add something along the lines of "slight changes in scaling factors (see methods) do not affect this finding."

Ok

Line 18 – the 476 ka – is this recalculated? I assume it is from the 2005 paper, but please state so.

This value is the value published in Kaplan et al. 2005, see table 2.

Line 24 before "Smedley" a word or two missing or open parentheses.

OK

p. 18. Top line 1 – spelling after the word Telken Line 27 grammar problem p. 19 Line 14-15. First two sentences - awkward writing.

Done

p. 20. per comment above, sorry, perhaps my non-expertise, but, can the authors say if there is a change in the U-Th data between Telken and Fenix moraines. A change in time of weathering rates? The relevance of the issue is testing the idea they discuss, in Kaplan et al. 2009. Is there a change in results and thus sediment transfer times/comminution age over the 1 Ma as valleys are excavated and Andes erode down?

There are changes in U-Th data as written in section 3.3. For 234U/238U these changes are small between Fenix and Telken compared to the total variability, however when combined to the variations of the f_alpha parameter, we observe a clear trend from Fenix to Telken in terms of age. The recycling time and weathering rate that we estimate from the Monte Carlo simulation is common to every samples. We just estimate that the weathering rate was higher before the final deposition (dated by 10Be exposure age) of the moraine. We cannot say whether there is a change in the transport time between 1 and 0 Ma, because this is a prerequisite from the Monte Carlo problem as we built it, that the recycling time is the same for every sample. This is an assumption that we make. The result that we obtain is an average. We clarified it in the paper.

Lines 17-19. Starting with "this is exacerbated. . ." I do not understand this sentence – it is not clear. The authors need to clarify.

This sentence has been removed since it was not really relevant and apparently not clear.

Line 23. What do they mean by 'moraine sediment.' That is very broad. Does this also apply to boulders? Or just find matrix in moraines? Please clarify (a similar question was raised earlier about the whole moraine being remobilized).

Ok, this is right, it remained unclear this question of size fraction of the moraine. Here we added '(the fine silty fraction)'.

p. 21. Line 24. I think I know what the authors mean, but as written, not recorded or not preserved is same thing. Rephrase to, not recorded because not preserved, or do they mean not preserved or ice did not reach this far? Something similar?

Yes, not recorded mean that ice did no reach this far (moraine not deposited in the area, but farther upstream). This is explianed in the following sentence. To make it clearer we linked both sentence with 'i.e. respectively'.

Table 1. It is becoming customary to also provide AMS ratios and carrier added (9Be). Given that (I think) it is standard enough to do this now in cosmo literature, that I have to insist these two columns be added to the table. Also, in caption, please remind reader of AMS standard used, given it is not common (except at ETH).

Added.

FIGURES Figure 1. It is difficult to evaluate fully the morphostratigraphic relations that are essential for interpretation of data from this figure: 1) need some photos associated with these maps – see comments above. This is especially so for the outwash linked to the moraine – cannot appreciate this from the map. Otherwise need to say perhaps not directly related to the specific

moraine – does it actually really matter for their main findings if they establish this? that is, deseado outwash is deseado outwash whether specifically linked to a specific crest in Singer et al, 2004. 2) I suggest one more panel really zooming in on the key area where the crests, outwash, and samples are. A panel D. This could be in the SOM if there is not enough room in the paper. The photos could also be in the SOM – just add one figure of several photos.. Any strat sections, with where the samples are from, that they can provide in SOM? I think it is always best to have the photos in the main text if there is space. However, if there is not enough space, at least to have them in the supplement info as a figure. 3) explain CDI and SPI in caption. SPI is obvious, but not CDI ? there are lots of glaciers in C. Darwin– what is specifically being highlighted and why for CD? the biggest ice mass in CD? 4) there is an error in the caption – the transect is in panel B (it says panel C, but this is in B)?. 5) after "[c]": add "[c] Inset shows a west-east transect. . ..." 6) For cosmo ages, it would better to make the symbols the same colors as those on figure 2. Easier for the reader not familiar with the area. I would also make the 2 outwash symbols as squares on Figure 1, just like on figure 2. 7) plot Deseado boulder that you refer to, from the literature (is from Kaplan et al. 2005 ?)

We improved the figure as per your recommendations. We also included pictures in supplementary materials. It is however difficult to see something, gven the fact that this area is rather flat. And it is hard to figure out the relief on those pictures. The boulders locations would add too much informations, and the figure would become unreadible.

Figure 2 1) maybe also plot here the Deseado boulder age (without erosion) that refer to? If do plot it, say it is a minimum age. Up to authors though whether they think it is best to plot the boulder cosmo age on figure 2, or it would be too messy. It would highlight again the moraine oulder ages too young. 2) Could all Deseado 2 ages be minima? Not just the 293? Deseado 1 has such a nice tight cluster, hard to envision being too minimum, but Deseado 2 is more scattered and only 2 ages. 3) I suggest showing the full LR 2005 curve. It is relatively easy to do, it does not take up much room and would be more informative than showing positive Stages. It would convey more info such as stage 8 not being a big global glacial maxima (cf., Hein et al.) 4) in caption, say what the effect is of using local PR versus global PR of Borchers et al, if significant ? see above comment?. Although not much for younger moraines, given the age equation is non-linear, it can add up for older Deseado moraines?

We imroved the figure as per your recommendations. Idem than for figure 1 with boulders data, we think the figure would become hardly readible.

Figure 3. 1) This plot is difficult to digest for the broader readership of ESD. Perhaps add some labels. For example, in the bottom 3 quadrants, can they add labels to all or some of the quadrants to summarize what we are looking at? In the top panels maybe also some more labels would help – what is the blue line? What are the red lines? It is explained in the caption, but, per my comment at top of review I am just trying to think of ways to make this figure more accessible to all readers. 2) Also, it is not clear – are these all hypothetical – or based on the monte carlo simulations – it is implied so ('results' mentioned in caption), but it is not specifically stated, and obvious to non-expert. 3) 4th line from bottom – 10Be ages – please explain which 10Be ages – reader cannot tell simply from the figure. 4) third line from bot- tom – "Weathering intensities are lower after moraine deposition" – how do you know this? That is, please explain from the plot how this is known. Or show visually with suggestion above to add labels.

The legend of this figure has been rewritten and we added a legend, and deleted a curve. Obviously this was unclear, we hope we made it clearer in the new legend.

Figure 6. Needed to explain a little better for non-experts – see comments at start of review. For non-experts, what is likelihood signify? Instead of Trecycle – spell out? Maybe a table in the SOM of all of these terms with a brief explanation, for non-experts? In panel A for example, they may add some info to the plot itself – some labels of what we are looking at. For example, what does the bend and dark vertical-aligned purple dots mean? Maybe a few labels on this plot and the bottom plot would make this more accessible to nonexperts in U-Th approach.

Lot of the informations are contained in the text (section 2.3.5). We tried to make clearer what the figure is. The 'bend and dark vertical-aligned purple dots' do not mean anything particular. What is really important is the likelihood of the dot. We rewrote the meaning of this term in the caption.

Supplement information.

Section 2. Given the prominent role the profiles and monte carlo simulation play in the paper - this section needs to be expanded to explain: 1) what the monte carlo optimization is and how it is done? Some of this is in the main text, but more details in the context of the Supplement are appropriate here in my opinion. 2) a paragraph (at least) explaining what we are looking at on Figures 1 and 2. What are these panels? The caption should refer to each panel. Or the text. Please explain each panel in more depth and/or in caption. The right panel – this is a key figure. Explain more – what is reader looking at? Not much is needed, perhaps just a longer caption, but more is warranted in explaining Figures 1 and 2 in the context of supplementary information and simply to describe what the reader is looking at, which are some of the key results discussed in the main text.

We added some sentences in the text to explain more about it.

Figure 3 – in caption, for non-expert, can they add one more sentence saying what f230 = 0 means? The authors also may want to add some labels along the lines of the comment above for main text figures.

Ok

Figure 5. Explain specifically where time comes from (e.g., U-Th dates associated with moraine/outwash dating. . ..and so on, see table 3)

Done

Per comments above, add photos and any strat sections. I would put the photos in the main text if there is room.

**Anonymous Referee #2**

This manuscript present some interesting cosmogenic and U-series isotope data. While the combination is interesting, it seems that isotopic systems are used inde- pendently with little overlap. Cosmogenic isotopes were used to date the moraines on the one hand, and U-series to infer information about sediment transport on the other hand. It almost reads like 2 independent manuscripts.

We feel the conmbined 10Be/U-Th data is key for our study. The 10Be data are used to date the moraines and U series are used to infer information on sediment transport. Both are not independant, because the ages determined using 10Be are needed to determine the sediment recycling time. Without these ages, the U-Th data do not give as much informations as they give with the 10Be data. Moreover, both the sequence of exposure ages and the recyling time determined from U-Th data, tell us that there is sediment reworked and at which timescale. Both set of data (10Be and U-Th) highlight the same timescale of 100 kyrs (recurrence of glacial cycles). So we think that it is appropriate to keep the two sets of data together.

While I command the authors' efforts, and this work should of course eventually be published, there are many aspects that the authors may want to consider, and which significantly affect how the U-series data are interpreted. I provide detailed comments in the PDF attached but the main points are that:

- the fractal correction is possibly not needed and result in an underestimation of recoil fractions. As a result, the authors explore a more complex (but less constrained) model (previously proposed by Scott et al. in the early 1990's, but also more recently re-introduced). A simple way to test this is to look at the type of isotherms obtained during BET analysis. Only a mesoporous would require the fractal correction to be invoked. I think Lee investigated this but as far as I know this is unpublished, but the authors may want to consider this point.

It is difficult to adress this comment since it is based on unpublished work. However, the isotherms we obtain are of type 2, following the classification of Brunauer (1945). Two extreme cases can be distinguished (see figure below), depending on their specific surfaces : Dubinin type (high surface → microporous), and BET type (lower surface → mesoporous). We report a plot showing the isotherms of our samples. One can see that they are all of the BET type (mesoporous). So if we follow the reasoning of the reviewer, a fractal correction must be applied to ours amples.

[Figure]

[Figure]

The comment of the reviewer made us realize there was a mistake in the section 1 of supplementary material. It is mentioned we applied the Dubinin-Radushkevich equation to the samples. A larger amount of samples (designed for ther purposes) were analyzed at the same time as those published in the paper. The section 1 of supplementary material was written by Thierry Reuschlé, who performed the analysis, for the whole group of samples, whereas only a few samples were concerned by the paper. The samples concerned by the Dubinin Radushkevitch equations (microporous) are actually not published in this paper. So we corrected this section, which now addresses the reviewer's concern.

- while the weathering model is presented, there is a lack of details on how it is solved, and what justifies the assumption made in the model (e.g. same 'recycling time' and weathering intensity for all samples? why? this record cover a very variable period of Earth's history). I only recommend rejection because I believe the extent of revisions to be undertaken is large, but a new manuscript should be submitted, taking into consideration the comments above and in the document attached.

We are aware that the assumptions made (same recycling time and weathering intensity for all samples) can be debated because the period covered by the record is the last several hundreds of throusands of years. Parameters that could drive variation of the weathering intensities are the climatic parameters mainly, and the freshness of the sediment. We consider a particular size fraction, so this latter parameter should not vary significanty. Climate varied, however the minima and maxima of the glacial interglacial variations remain almost similar from a cycle to another by 700-800 ka (Mid Pleistocene Transition). We can then assume that the processes occuring during a glacial cycle are comparable in intensity from a cycle to another.

Regarding the recycling time, the main other parameter (apart from climate) influencing it is the morphology (mainly valleys and overdeepening). The Great Patagonian Glaciation occured before 1 Ma, these glaciations are most probably responsible for carving the U shaped valleys

and the overdeepening in which the lake is nowadays. Moreover the results of the numerical experiments of Kaplan (2009) about the extent of ice related to the elevation of the accumulation area and the depth of the overdeepening suggest that this overdeepening was most probably already carved before 1 Ma. Recently Christeleit (2017) published a paper in which she argues that much of the observed valley relief was carved between 10 and 5 Ma. So it seems reasonable to assume that the recycling time remained constant over the period covered by the samples. Fianlly, the 10Be data suggest the same timescale than U-Th suggest : apparently every second glacial cycle is preserved n the morainic sequence, suggesting a timescale of 100 to 200 ka of sediment reworking. Of course this must have changed on longer timescale, but over the period covered by the moraine sequence it seems those variations are not detectable.

We recognize those points were not clear, or absent in the paper, we improved it by adding a paragraph in section 2.3.5.

Please also note the supplement to this comment: https://www.earth-surf-dynam-discuss.net/esurf-2017-45/esurf-2017-45-RC2- supplement.pdf

Page 8 : in theory this is correct. but this ignores that 234U will be in damaged sites, so f230 is likely to be greater than the value given in equation 1.

This is right, it remains however impossible to quantify this. We added a sentence to make ths point.

not new. already given in Scott et al. 1992, and Dosseto and Schaller 2016.

We did not find Scott et al. 1992 : is it « Natural decay series studies of the redox front system in the Pocos de Caldas uranium mineralization » by MacKenzie and Scott et al. (1992) ? We added the reference Dosseto and Schaller.

Page 9 : problem is that Lee shows that ignition affects surface area. also, Martin et al. 2015 show that ignition is not optimal for (234U/238U)

Lee showed slight differences on 234U/238U between the ignition method and the H2O2 leach for removing organic matter. However other studies showed that H2O2 leaches were not fully selective (Tessier 1979). This explained in point #2 of section 2.3.2.

why introduce a bias by removing the <4um fraction? Handley shows that it is not recommended

If clays are directly precipitated onto the mineral surface after weathering then they should be preserved and analyzed. In such a case the problem and equation do not need to take weathering into account. But how can we be sure that clays present in the sample are kept on mineral surface? Moreover our understanding of Handley's work is that she points out that some clays should be conserved and some should be removed, but it is impossible to isolate both. If clay fraction is made by primary minerals smaller than 4 $\mu$m then they should be conserved. However our MEB images show that clays (secondary minerals) are present.

Page 10 : sodium citrate should be added at each step to avoid re-adsorption. while not affecting the (234U/238U) ratio (see Martin et al. 2015) this could be critical for Th and could explain

differences in Th concentration (although I acknowledge it doesn't seem to affect the 230Th/238U ratio)

We acknowledge this would have been better to add this sodium citrate. As mentionned the problem is for Th. That's why we tested the sensitivity of the Monte Carlo to changing values of fa230 (recoil loss fraction of 230Th) : a lower fa230 is equivalent to readsorption of the ejected 230Th. See supplementary material. Also, increasing the number of chemicals added to the samples increases the chance of disolving or corroding the surface of the grains.

Page 11 : results on rock standards need to be reported.

Page 14 : no evidence there was anything to remove in the first place

Page 15 : clays are not a problem: the are the product of incongruent weathering so unless they are completely precipitated from soluion (which is only common in marine environments; see textbooks on clays), they are not a concern and shouldn't be removed,

In the lake clays can be precipitated. So we cannot be sure that the clays in the samples are directly originated from the weathering of the silts that we analyze.

Page 16 : or the recoil fraction is under-estimated because applying a fractal correction when it's not needed. Depending on the shape of adsorption-desorption isotherms, you can identify the type of surface porosity; and the fractal correction should allow be applied when dealing with a mesoporous material, because only then the size of surface pores is comparable to the length of recoil and including these pores would result in overestimation of the surface area and recoil fraction (and thus requires the fractal correction). if isotherms suggest a microporous or macroporous material (as it is often the case), no fractal correction is needed. In this case, you find that you can obtain numerical values for the comminution age. For instance, D1-U-M yields 400 ka; D2-U-M gives 320 ka.

See our response above to your main comments.

does this assume that all samples have the same 'recycling time'? what would justify such assumption? so you have 8 samples x 2 ratios = 16 inputs to the moded; and 5 outputs (again assuming the recycling time is the same for all samples?)? in this case, the number of unknowns could be increased since they're so many inputs. for instance, samples from different moraines could have different recycling times (thus, 13 outputs). I think this needs a bit more details (even if the model is described above) before jumping into the results.

See our response above to your main comments.

---

## Author Response (AR2)

Dear Associate Editor,

We thank you for your consideration of the manuscript and the positive answer, and we thank again the reviewers who allowed the paper to improve considerably.

We address below the different points that you mentionned in your decision letter.
We hope these changes will help.

Best regards
Antoine Cogez

**Associate Editor Decision: Publish subject to minor revisions (review by editor)** (04 Jan 2018) by Jane Willenbring

Comments to the Author:

Thank you for your submission of this work. I have considered excellent and thorough reviews given by the two anonymous reviewers. It seems that most of the comments were addressed and incorporated into the manuscript.

I noticed that there were still no rock standards published along with the U-Th data even though this was requested by reviewer #2

Ok, we forgot it. We added the data in table 3 for a rock standard analyzed following the same protocol as the samples (except the leaches) and a reference to it in section 2.3.3. We would like to point out that the different duplicates that we also show in this table are more relevant for the scope of this study (reliability of the leaching protocole and the silicate digestion).

and there is an uncited reference listed:
Prytulak, J. et al. (2008). An inter-laboratory assessment of the thorium isotopic composition of synthetic and rock reference materials. Geostandards and Geoanalytical Research, 32, 65–91. Perhaps this was an omission.

It seems this reference is actually Sims et al. In the references Prytulak is the last author listed (line 15 of page 30) in the long list of authors of this paper. It appears shifted from the first line of the reference (line 14 of page 30, not 15) mentionning Sims… etc

Figure caption 1: dureation should be "duration" and vetical should be "vertical"
In fact, this whole sentence: "i.e. between Y axis and this dashed vetical line. (lower panels: d, e and f) the co-evolution of these ratios presented above." is not proper English. Please rephrase the punctuation and/or text to make it make more sense.
Also in the Fig 1 caption: "In order to make sense of the complexity of the different possible patterns in the framework of the comminution model incorporating weathering (see section 2.3.1. Different cases are plotted, corresponding to different f values, and different weathering intensities (see legend)." This is not a sentence and doesn't make sense. Please revise.

We changed those sentences to make them more understandable.
« Our 10Be exposure ages give the time elapsed since this event (shown by the dashed vertical line), whereas the U-Th data indicate the duration of the first phase since comminution and remobilization (from the x-axis origin to the dashed vertical line). (Lower panels: d, e and f) The co-evolution of the ratios in a, b and c, respectively. Different scenarios are plotted corresponding to different f-values and weathering intensities to illustrate potential patterns in the comminution weathering model (see legend and Section 2.3.1). »

There are some formatting issues with the references in the document.

Fixed, as exhaustively as possible.

Other than these easy adjustments and additions, the manuscript will be ready to publish.

---

## Author Response (AR3)

Paris, February 7th 2018

Antoine Cogez
Antoine.cogez@gmail.com
Institut de Physique du Globe de Paris
1 rue Jussieu
75005 Paris

Dear Editors,

We thank you for your kind attention reviewing and commenting the manuscript, which helped improving the manuscript. We took into account your last comments. Regarding the point about weathering in the comminution age theory, we acknowledge your comment. We tried to rephrase some sentences in the introduction and the abstract to soften the language, and included the reference of Andersen et al. (2013), that we did not cited yet.

Best regards
Antoine Cogez